# PEARL: TOWARDS PERMUTATION-RESILIENT LLMS

**Liang Chen**[1]   **Li Shen**[2*]   **Yang Deng**[3]   **Xiaoyan Zhao**[1]   **Bin Liang**[1]   **Kam-Fai Wong**[1*]
[1]The Chinese University of Hong Kong  [2]Shenzhen Campus of Sun Yat-sen University  [3]SMU
{lchen, kfwong}@se.cuhk.edu.hk  mathshenli@gmail.com

## ABSTRACT

The in-context learning (ICL) capability of large language models (LLMs) enables them to perform challenging tasks using provided demonstrations. However, ICL is highly sensitive to the ordering of demonstrations, leading to instability in predictions. This paper shows that this vulnerability can be exploited to design a natural attack—difficult for model providers to detect—that achieves nearly 80% success rate on LLaMA-3 by simply permuting the demonstrations. Existing mitigation methods primarily rely on post-processing and fail to enhance the model's inherent robustness to input permutations, raising concerns about safety and reliability of LLMs. To address this issue, we propose **Pe**rmutation-**r**esilient **l**earning (**PEARL**), a novel framework based on distributionally robust optimization (DRO), which optimizes model performance against the worst-case input permutation. Specifically, PEARL consists of a permutation-proposal network (P-Net) and the LLM. The P-Net generates the most challenging permutations by treating it as an optimal transport problem, which is solved using an entropy-constrained Sinkhorn algorithm. Through minimax optimization, the P-Net and the LLM iteratively optimize against each other, progressively improving the LLM's robustness. Experiments on synthetic pre-training and real-world instruction tuning tasks demonstrate that PEARL effectively mitigates permutation attacks and enhances performance. Notably, despite being trained on fewer shots and shorter contexts, PEARL achieves performance gains of up to 40% when scaled to *many-shot* and *long-context* scenarios, highlighting its efficiency and generalization capabilities.

## 1 INTRODUCTION

A hallmark of human intelligence is the ability to learn and execute new tasks by reasoning from a few examples. Mirroring this, in-context learning (Brown et al., 2020), as a crucial supplement to zero-shot prompting, has shown promising results across a spectrum of complex tasks (Cobbe et al., 2021; Chen et al., 2023a; OpenAI et al., 2023). Despite these advancements, the in-context learning capabilities of LLMs remain fragile. LLMs exhibit sensitivity to permutations of provided demonstrations (Lu et al., 2022; Zhao et al., 2021; Reynolds & McDonell, 2021), posing challenges for prompt engineering and leaving a significant gap in achieving human-like adaptability.

Most existing studies on ICL primarily aim to enhance normal-case performance (Min et al., 2022; Wei et al., 2023), with limited attention to improving permutation robustness. Current approaches to addressing this issue typically involve modifying training objectives to mitigate the limitations of transformers' unidirectional attention (Xiang et al., 2024) or designing permutation-equivariant architectures (Chen et al., 2023b). However, these methods often lack scalability. Alternatively, decoding-stage techniques such as output calibration (Zhao et al., 2021) and demonstration order optimization (Lu et al., 2022) introduce additional computational overhead per inference call, further limiting their practicality. Thus, a critical need remains for methods that fundamentally enhance LLMs' inherent robustness to input permutations.

In this work, we conduct extensive experiments on LLaMA-3, an advanced open-source LLM, to reassess its vulnerability to permutation-based attacks from an adversarial perspective (§2). Our findings reveal that even LLaMA-3 remains highly susceptible to simple attacks that alter only

---

*Corresponding Authors.

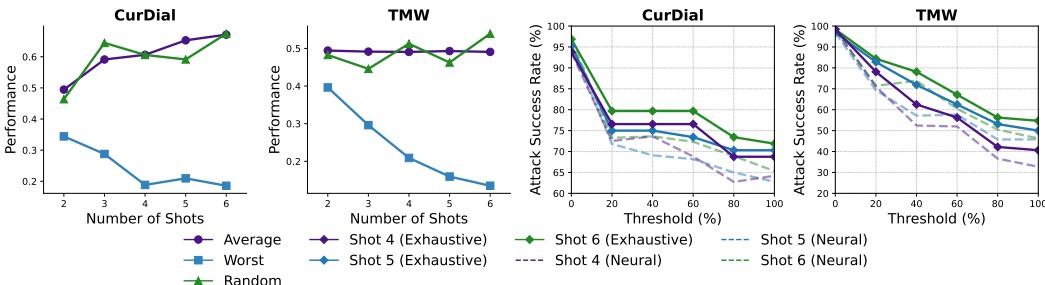

Figure 1: Performance and attack success rates of Llama-3 on CurDial and TMW datasets. Left panels: Random, average and worst-case performance as a function of shot number. Right panels: Attack success rates for exhaustive and neural search attack methods at different thresholds.

the order of ICL demonstrations. These attacks preserve the semantic content of examples and introduce no adversarial modifications, yet they degrade performance with success rates exceeding 80%. Consequently, they are difficult for model providers to detect but significantly undermine LLM performance, highlighting a critical vulnerability concern.

To counteract the vulnerability to input permutations, we introduce a novel **Pe**rmutation-**r**esilient **l**earning (PEARL) framework, which is based on distributionally robust optimization (DRO) (Ben-Tal et al., 2011). Unlike standard empirical risk minimization training, adopted by most supervised fine-tuning (SFT) methods, which views each training instance merely in terms of its one or several permutations observed during training, DRO conceptualizes each instance as part of a broader distribution that includes all conceivable permutations. This comprehensive set of all possible permutations is termed the ambiguity set. By explicitly identifying and optimizing the worst-case within this ambiguity set, our strategy substantially enhances the resilience of LLMs against all different permutations. This paradigm shift—from considering training instances as single data points to viewing them within a distribution of potential permutations—equips the model to better prepare for and generalize to combinatorial input scenarios.

Specifically, PEARL operationalizes DRO as a two-player game, consisting of a permutation-proposal network (P-Net) as the adversary and the LLM as the target model. For each training instance, P-Net identifies a challenging permutation of given demonstrations, aiming to maximize the LLM's loss. Conversely, the LLM strives to minimize its loss under the P-Net's manipulation, thereby performing well on these difficult examples. P-Net treats the generation of the adversarial ICL permutation as an optimal transport (OT) (Monge, 1781) problem between the distribution over input permutations and the distribution of challenging permutations for LLMs. We solve the OT problem using the Sinkhorn algorithm (Sinkhorn, 1966) with an element-wise entropy constraint designed to prevent trivial solutions. Through adversarial training (AT), both networks improve iteratively. Ideally, at convergence, the P-Net represents a uniform distribution across all permutations, as the LLM handles all possible permutations equally well.

We validate our method in two scenarios: (1) pretraining a transformer to in-context learn linear functions (Garg et al., 2022), and (2) instruction tuning of LLMs on the Super-Natural Instructions (Wang et al., 2022). The results demonstrate that, on unseen tasks, our method consistently improves both the average and worst-case performance of LLMs across different permutations, effectively defending against permutation-based attacks. Furthermore, despite being trained with much smaller configurations, our method generalizes effectively to ***many-shot ICL*** and ***long sequences***, achieving performance gains of 24% to 40%. These results highlight the efficiency and generalization capabilities of our approach. The code is available at `https://github.com/ChanLiang/PEARL`.

## 2 REVISITING PERMUTATION VULNERABILITY IN LLMS

This section investigates performance fluctuations in SOTA open-source LLMs when presented with different permutations of given demonstrations. Additionally, from an adversarial perspective, we explore whether this vulnerability can be exploited to devise an effective attack on LLMs.

**Experimental Setups**    To conduct evaluations, we select two tasks from Super-NaturalInstructions (Wang et al., 2022), including Curiosity-based Dialog (CurDial) and TellMeWhy QA (TMW). We test 100 samples for each task, with each sample structured as a quadruple consisting of (instruction, demonstrations, input, output). The number of demonstrations (shots) ranges from two to six. Following Wang et al. (2022), the performance is measured using the ROUGE-L (Lin, 2004). We analyze the permutation vulnerability of LLaMA-3-8B on two settings as follows:

**1) Permutation Vulnerability on Different Number of Demonstrations**    We first examine the average and worst-case performance of the model across different permutations of input demonstrations and the effect of scaling the number of demonstrations. As shown in the left of Figure 1, there is a notable observation: ***adding demonstrations is a double-edged sword.*** Increasing the number of demonstrations (*shots*) generally enhances the model's average performance due to richer contextual information. However, it can simultaneously worsen the worst-case performance. This suggests that while additional demonstrations provide useful context, the exponentially growing number of permutations (n!) increases the risk of the model performing poorly on certain input configurations.

**2) Input Permutation as Attack**    We then consider a two-party adversarial scenario between a malicious user (attacker) and a model provider (defender). The attacker aims to induce compromised responses from LLMs by permuting ICL demonstrations, making the attack less detectable. Given a task $D = \{(p_i, x_i, y_i)\}$, a sample is successfully attacked if its relative performance degradation, induced by the attacker, exceeds a threshold $\delta \in [0\%, 100\%]$. Here, $p_i$ is an ICL prompt with $n$ demonstrations. The set of all possible demonstration permutations is $\mathbb{P} = \{\Pi_0, \ldots, \Pi_{n!-1}\}$, where $|\mathbb{P}| = n!$. Let $g$ be a performance metric (e.g., ROUGE-L). The attack success rate (ASR) for task $D$ is defined as:

$$\text{ASR}(D, \delta) = \frac{1}{|D|} \sum_{i=1}^{|D|} \mathbb{I}\big((\mu_i - \omega_i)/\mu_i \geq \delta\big) \tag{1}$$

where $\mathbb{I}$ denotes the indicator function, $|D|$ is the size of the dataset, and $\delta$ is the threshold. The average performance of the i-th sample, $\mu_i$, is defined by:

$$\mu_i = \mathbb{E}_{\Pi \sim \mathbb{P}}[g(\Pi \cdot p_i, x_i; y_i)] = \frac{1}{n!} \sum_{j=1}^{n!} g(\Pi_j \cdot p_i, x_i; y_i) \tag{2}$$

and $\omega_i$ is the compromised performance induced by the attack strategy adopted by the malicious user. Here, we analyze two attack method:

- **Exhaustive Search Attack**: To *calculate the upper bound* of the effect the permutation-based attack can achieve, we assume that the malicious user has unlimited attempts and conducts an exhaustive search. For each sample $(p_i, x_i, y_i)$, this process involved testing all possible permutations of demonstrations in $Q_i$ and identifying the permutation that yields the poorest performance. In this case, the attacked performance is calculated as follows:

$$\omega_i = \min_{\Pi \in \mathbb{P}} g(\Pi \cdot p_i, x_i; y_i) \tag{3}$$

- **Neural Search Attack**: To *approximate the upper bound* established by the exhaustive search when the number of attempts is limited, we employ a meta-learning approach to optimize a permutation-proposal network (P-Net). As illustrated in Figure 3 (details are in the Methods section), during training, this network takes the standard sample $(p_i, x_i, y_i)$ as input and outputs a permutation matrix $\Pi_i$. The permuted samples $(\Pi_i \cdot p_i, x_i, y_i)$ are then fed into the LM to maximize its loss function. During testing, the network generates the most challenging permutation $\Pi_i$ for each sample $(p_i, x_i, y_i)$. Then the attacked performance is calculated as follows:

$$\omega_i = g(\Pi_i \cdot p_i, x_i; y_i), \qquad \text{s.t. } \Pi_i \sim \text{P-Net}(p_i, x_i, y_i) \tag{4}$$

As shown in the right of Figure 1, the results indicate that ***permutation attacks are effective and approachable.*** Leveraging this characteristic, the exhaustive search attack successfully attacks over 50% and 80% of the samples with $\delta = 50\%$ on two datasets respectively, and the neural attack achieved a successful rate close to this upper bound across different shots. These results demonstrate that this vulnerability poses a real concern, even for advanced LLMs like LLaMA-3.

**Remark**    These deficiencies may directly stem from the fundamental limitations of standard Empirical Risk Minimization (ERM) training, which focuses on optimizing average performance while neglecting worst-case performance. We discuss this issue in depth in the next section and propose a method to address the model's improper behavior on unseen but practically valid input spaces.

## 3    PERMUTATION-RESILIENT LEARNING (PEARL)

### 3.1    INSTRUCTION TUNING VIA DRO

Our objective is to train a LLM to perform well across all possible permutations of given demonstrations when prompted with few-shot instructions.

In supervised fine-tuning for few-shot learning, the LLM is trained to predict an output $y \in \mathcal{Y}$ given an input $x \in \mathcal{X}$ and a few-shot instruction $p \in \mathcal{P}$, where $p$ typically consists of a sequence of demonstrations, each being an input-output pair. Let $\Theta$ denote the parameter space of the language model, and let $\ell : \Theta \times (\mathcal{P} \times \mathcal{X} \times \mathcal{Y}) \to \mathbb{R}_+$ be a nonnegative loss function measuring the discrepancy between the model's prediction and the true output. The standard approach is to find parameters $\theta \in \Theta$ that minimize the empirical loss over the training data via empirical risk minimization:

$$\hat{\theta}_{\text{ERM}} := \arg\min_{\theta \in \Theta} \mathbb{E}_{(p,x,y) \sim \hat{P}}[\ell(\theta; p, x, y)] \tag{5}$$

where $\hat{P}$ denotes the empirical distribution derived from the training dataset.

Under appropriate assumptions, learning theory (Vapnik, 1999; Shalev-Shwartz & Ben-David, 2014) guarantees that models trained via ERM perform well on the test distribution given sufficient training data. However, in practice, models trained using ERM often fail to generalize well to different permutations of the same set of demonstrations. This occurs because the training set covers only a subset of all possible permutations of the demonstrations, and during testing, the model may encounter permutations not seen during training, leading to a significant degradation in performance.

To systematically address the permutation sensitivity issue, we propose fine-tuning under the framework of distributionally robust optimization, which optimizes the risk under the worst-case distribution within a specified ambiguity set. Specifically, we aim to solve:

$$\hat{\theta}_{\text{DRO}} := \arg\min_{\theta \in \Theta} \left\{ \sup_{Q_\Pi \in \mathcal{Q}} \mathbb{E}_{(p,x,y) \sim Q_\Pi}[\ell(\theta; p, x, y)] \right\} \tag{6}$$

The ambiguity set $\mathcal{Q}$ is constructed as the convex hull of all distributions obtained by permuting the prompts in the empirical distribution $\hat{P}$. Specifically, we define:

$$\mathcal{Q} := \left\{ \sum_{\Pi \in \mathbb{P}} q_\Pi \, Q_\Pi \, \middle| \, q \in \Delta_{|\mathbb{P}|-1} \right\}, \quad \text{where} \quad Q_\Pi := \left\{ (\Pi \cdot p, \, x, \, y) \, \middle| \, (p, x, y) \sim \hat{P} \right\}. \tag{7}$$

Here, $\Pi$ is a permutation matrix that reorders the sequence of demonstrations in $p$, and $\mathbb{P}$ denotes the set of all such matrices. The vector $q$ lies in the $|\mathbb{P}| - 1$-dimensional probability simplex $\Delta_{|\mathbb{P}|-1}$.

To illustrate the advantages of DRO over ERM in handling different permutations, consider the example in Figure 2. For a 3-shot training example $(p, x, y)$ with prompt $p$ containing three demonstrations, there are six possible permutations denoted as $(p^0, x, y), \ldots, (p^5, x, y)$, indexed from 0 to 5. $\hat{P}$ denotes the empirical distribution of permutations in training data, represented by blue bars. The bars show that permutation indices 0, 1, and 4 appear in training data with frequencies, while permutations 2, 3, and 5 do not appear. $P_\theta$ represents the distribution learned by the LLM, represented by purple curves. In panel (a), the ERM-trained model assigns higher probabilities to frequently occurring permutations $(0, 1, 4)$

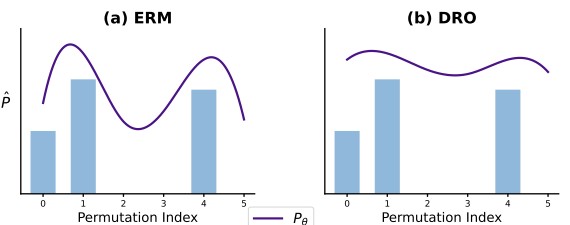

Figure 2: Comparison of models trained under ERM and DRO paradigms. The blue bars represent the empirical distribution $\hat{P}$ of training data, showing different frequencies of six permutations in the training set. The purple curves denote the learned distribution $P_\theta$ by (a) ERM and (b) DRO models, illustrating their different behaviors on less appeared but valid permutations.

and lower probabilities to less frequent ones $(2, 3, 5)$, leading to poor performance on unseen permutations during testing. In contrast, panel (b) shows that the DRO-trained model distributes probabilities more uniformly across all possible permutations, as it explicitly considers them all (Equation (6)) during learning. This demonstrates how DRO mitigates ERM's limitations by encouraging models to assign reasonable probabilities to all valid permutations, regardless of their frequency in training data.

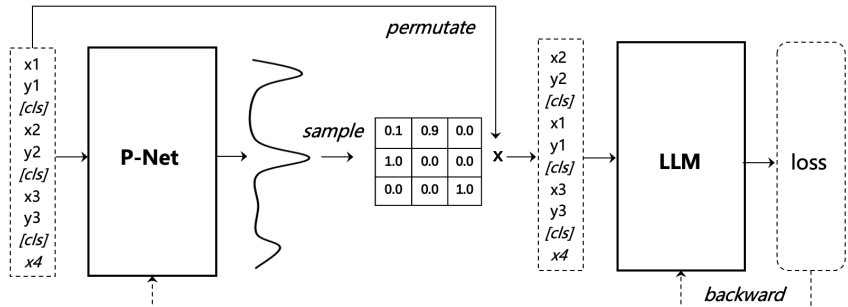

Figure 3: An overview of the learning framework. The P-Net is a small model incorporating the Sinkhorn operator, trained jointly with the LLM under the adversarial optimization algorithm. *Note that the permutation matrix operates on the input sequence's embeddings (simplified here as text sequences for clarity).* After training, only the LLM is retained while the P-Net is discarded.

## 3.2 LEARNING TO GENERATE PERMUTATIONS VIA P-NET

To enable our DRO framework to function effectively, we need to efficiently find the worst-case scenario within the ambiguity set (solve the max step in Equation (6)). Directly addressing this problem through exhaustive search is computationally infeasible due to the exponential search space.

We address this challenge by drawing inspiration from optimal transport, treating the problem as transportation between permutation distributions. We introduce the **P**ermutation-**p**roposal **Net**work (**P-Net**), denoted as P-Net: $(\mathcal{P} \times \mathcal{X} \times \mathcal{Y}) \rightarrow \Delta(\Pi)$, which learns a distribution over permutations to increase task difficulty for the LLM given input examples. As shown in Figure 3, we sample challenging permutations from this distribution to reorder the given demonstrations.

Specifically, P-Net consists of two components: a parameter part that extracts features and models the relationships between demonstrations, a non-parameter part using the Sinkhorn algorithm to build the distribution $\Delta(\Pi)$, and Gumbel sampling for differentiable sampling from it ($\Pi \sim \Delta(\Pi)$).

**Parameter component.** The parameter component consists of a *feature extractor* and a *cross-relationship modeling layer*. The feature extractor is an encoder model that takes an ICL prompt composed of $n$ demonstration pairs $p = \{(x_i, y_i)\}_{i=1}^n$ and a predicting sample $(x, y)$, and produces their representations as follows:

$$([\text{CLS}], (x_1, y_1), \ldots, [\text{CLS}], (x_n, y_n), [\text{CLS}], (x, y)) \xrightarrow{\text{Encoder}} (\mathbf{h}_1, \mathbf{h}_2, \ldots, \mathbf{h}_n, \mathbf{h}_{n+1}), \quad (8)$$

where $\mathbf{h}_i$ is the representation corresponding to the $i$-th [CLS] token, which is often used to segment and extract the representation of sequences (Devlin et al., 2019b; Lu et al., 2021).

After extracting the representations of $n$ demonstrations, we have $H = (\mathbf{h}_1, \mathbf{h}_2, \ldots, \mathbf{h}_n) \in \mathbb{R}^{n \times h}$. We then model the pairwise relationships among the demonstrations. Specifically, we design a simple *cross-demonstration layer* to obtain a relationship matrix $\mathbf{R} \in \mathbb{R}^{n \times n}$ that captures the pairwise relationships between each pair of demonstrations, defined as:

$$\mathbf{R} = g\left(HWH^\top\right), \quad (9)$$

where $W \in \mathbb{R}^{h \times h}$ is a weight matrix, and $g$ denotes a nonlinear activation function.

The matrix $\mathbf{R}$ can be interpreted as an **adjacency matrix** in graph theory, where demonstrations serve as nodes, and $R_{ij}$ represents the relationship between demonstrations $i$ and $j$. Specifically, we define $R_{ij}$ as the potential increase in task difficulty for the LLM if demonstrations $i$ and $j$ are swapped; higher values of $R_{ij}$ indicate that swapping these two demonstrations may significantly impact prediction. Thus, this parameterized component models an **edge prediction** process.

However, while $\mathbf{R}$ captures the potential for swapping between demonstrations, it is not yet suitable for sampling permutations because its elements can take any real values and do not necessarily form a valid probability distribution. To convert $\mathbf{R}$ into a distribution over permutations $\Delta(\Pi)$ that we can sample from, we introduce a non-parameter component.

**Non-parameter component.** The non-parameter component aims to transform the adjacency matrix $\mathbf{R}$ into a *doubly stochastic matrix*, representing a probability distribution over permutations. Specifically, following Adams & Zemel (2011); Mena et al. (2018), we adopt the Sinkhorn operator $S(\cdot)$ to obtain such matrices through an iterative process of row and column normalization:

$$S(R) = \lim_{l \to \infty} \left( \mathcal{T}_c \left( \mathcal{T}_r \left( \exp(R) \right) \right) \right), \tag{10}$$

$$\mathcal{T}_r(R) = R \oslash \left( R \mathbf{1}_n \mathbf{1}_n^\top \right), \quad \mathcal{T}_c(R) = R \oslash \left( \mathbf{1}_n \mathbf{1}_n^\top R \right), \tag{11}$$

where $\mathcal{T}_r(R)$ and $\mathcal{T}_c(R)$ represent the row and column normalization operators, respectively; $\oslash$ indicates element-wise division; and $\mathbf{1}_n$ is a column vector of ones. As established by (Sinkhorn, 1966), the Sinkhorn operator $S(R)$ strictly converges to a doubly stochastic matrix as the number of iterations $l$ approaches infinity.

To ensure a differentiable process when sampling permutations from the distribution, the Gumbel trick (Jang et al., 2017) is applied:

$$\Pi = \lim_{\tau \to 0} S\left( (R + G)/\tau \right), \tag{12}$$

$$G_{ij} = -\log\left( -\log G'_{ij} \right), \quad G'_{ij} \sim U(0, 1), \tag{13}$$

Where $G \in \mathbb{R}^{n \times n}$ is the Gumbel noise and $\tau$ the temperature. As $\tau$ approaches zero, the result approximates a permutation matrix $\Pi$. Hyperparameters are studied in Appendix C.

By regarding permutation generation as an optimal transport problem and implementing it through P-Net, we transform the input permutation distribution into a target distribution. Next, we introduce how P-Net is co-optimized with the LLM to make the target permutation distribution the most challenging for the current LLM.

## 3.3 ADVERSARIAL OPTIMIZATION

As illustrated in Figure 3, we adopt an adversarial optimization framework to jointly train the LLM and the P-Net. Let $\theta$ and $\phi$ denote the parameters of the LLM and P-Net, respectively. For each sample $(p, x, y)$ drawn from the empirical distribution $\hat{P}$, the P-Net generates an adversarial permutation $\Pi$ that maximizes the LLM's loss. In response, the LLM aims to minimize its loss, adversarially influenced by the P-Net. The LLM's loss function is defined as:

$$L_{\text{lm}}(\phi; \theta) = \mathbb{E}_{(p,x,y) \sim \hat{P}, \Pi \sim \text{P-Net}(\phi;p,x,y)}[\ell(\theta; (\Pi \cdot p, x, y))] \tag{14}$$

To prevent the P-Net from collapsing to trivial solutions, such as producing uniform permutations that degrade demonstration semantics, we introduce an entropy-based regularization term:

$$L_{\text{ent}}(\phi) = \mathbb{E}_{(p,x,y) \sim \hat{P}, \Pi \sim \text{P-Net}(\phi;p,x,y)}[\mathcal{H}(\Pi)], \tag{15}$$

where $\mathcal{H}(\cdot)$ denotes the element-wise entropy function.

This results in a two-player min-max optimization problem with the following objective:

$$\min_{\theta} \max_{\phi} \left( L_{\text{lm}}(\phi; \theta) - \beta L_{\text{ent}}(\phi) \right), \tag{16}$$

where $\beta$ is a hyperparameter controlling the strength of entropy regularization.

We employ alternating optimization to iteratively update $\theta$ and $\phi$. The full training procedure is detailed in Algorithm 1.

---

**Algorithm 1:** Adversarial Optimization Algorithm for PEARL

**Input:** $\theta, \phi$ (LLM, P-Net); $\eta_\theta, \eta_\phi$ (learning rates); $m$ (inner steps); $\beta$ (entropy coefficient)
**repeat**
    **for** $t = 1$ **to** $m$ **do**
        $(p, x, y) \sim \hat{P}$ ;                    // Sample training examples
        $\Pi \sim \text{P-Net}(\phi, p, x, y)$ ;           // Generate permutations
        $L_{\text{lm}}(\phi, \theta) \leftarrow \ell(\theta; \Pi \cdot p, x, y)$ ;        // Compute LLM loss
        $L_{\text{ent}}(\phi) \leftarrow \mathcal{H}(\Pi)$ ;        // Compute entropy regularization
        $\phi \leftarrow \phi + \eta_\phi \nabla_\phi(L_{\text{lm}} - \beta L_{\text{ent}})$ ;        // Update P-Net
    **end**
    $\theta \leftarrow \theta - \eta_\theta \nabla_\theta L_{\text{lm}}(\phi, \theta)$ ;        // Update LLM
**until** *convergence*;

---

## 4 IN-CONTEXT LEARNING WITH LINEAR FUNCTIONS

### 4.1 DATASETS AND EVALUATION METRICS

We investigate in-context learning on linear functions $f(x) = w^\top x$, where $w \in \mathbb{R}^d$, following Garg et al. (2022); Guo et al. (2024b). For each $w$, we construct each example $p^i = (x_1, f(x_1), \ldots, x_i, f(x_i), x_{i+1})$ containing $i$ input-output demonstration pairs and a query input $x_{i+1}$. A language model $LM$ is trained to minimize:

$$\min_\theta \mathbb{E}_p \left[ \frac{1}{k+1} \sum_{i=0}^k \ell(\theta; p^i, f(x_{i+1})) \right], \tag{17}$$

where $\ell(\cdot)$ is the MSE loss and $k$ is the number of demonstrations. During testing, we evaluated performance using the same MSE metric. We report the normalized squared error $((LM(p) - w^\top x_{\text{query}})^2/d)$, where $d$ is the problem dimension. Detailed settings are in Appendix A.1.

### 4.2 IMPLEMENTATION DETAILS AND BASELINES

**Architecture and Training.** We implement $L_\theta$ using a GPT-2 base model (Radford et al., 2019) and train it from scratch on a generated dataset using the AdamW (Loshchilov & Hutter, 2019). Key training parameters include a batch size of 128 and 500k training steps. In the PEARL framework, the P-Net is initialized as a BERT-base (Devlin et al., 2019a) and also trained from scratch. Implementation details are in Appendix A.2.

**Baselines**. Consistent with Garg et al. (2022), we adopt an empirical risk minimization method with curriculum learning (Bengio et al., 2009; Wu et al., 2020) **(ERM+CL)** to train the model. The training process gradually increase the number of demonstrations presented to the model, allowing for progressive learning of more complex patterns and making the training more stable.

### 4.3 EVALUATION RESULTS

We evaluate the effect of permutations on the worst-case and average performance of different methods, as well as each method's defence capability against permutation attacks.

Table 1: Normalized MSE across permutations.

| Shot | Method | Avg. | Worst. |
|------|--------|------|--------|
| 3 | ERM+CL | 1.45 | 2.67 |
|   | PEARL | **0.86** (+40.7) | **0.92** (+65.5) |
| 4 | ERM+CL | 1.20 | 3.34 |
|   | PEARL | **0.79** (+34.1) | **1.11** (+66.8) |
| 5 | ERM+CL | 1.28 | 5.03 |
|   | PEARL | **0.87** (+32.0) | **1.33** (+73.6) |

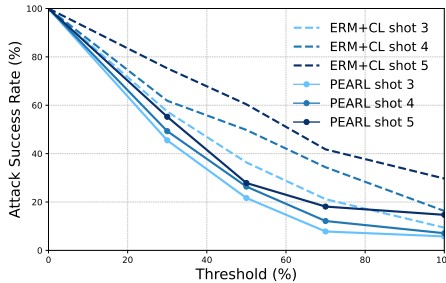

Figure 4: Comparison of attack success rates.

As shown in Table 1, the performance gap between average and worst-case performance across permutations for the baseline methods was significant, indicating substantial vulnerability to permutations. Specifically, the worst-case performance of the baseline methods decreased dramatically compared to their average performance, with the relative performance drop increasing from 74.6% at 3 shots to 84.1% at 4 shots, effectively losing most of the performance gains achieved by increasing the number of shots. In contrast, our method, PEARL, not only improved the average performance but also significantly enhanced the worst-case generalization performance compared to the baselines. While the average performance gains tend to plateau as the number of shots increases, the worst-case performance gains continue to rise, increasing from 65.5% at 3 shots to 73.6% at 5 shots.

Figure 4 depicts the proportion of successfully attacked samples in terms of (1) different attack success thresholds and (2) number of demonstrations (shots). The former considers more pessimistic scenarios (attacked samples drop a large margin), while the latter examines larger input spaces. We observed that PEARL's advantage increased as the threshold grew. At $\delta > 50\%$, the defence success

rate for PEARL across all shots was approximately double that of the baseline methods. This indicates that PEARL can effectively prevent pessimistic scenarios (samples attacked with a large threshold). Moreover, PEARL's performance improved with an increasing number of shots, suggesting better scalability compared to baseline methods.

## 5 INSTRUCTION FINE-TUNING OF LARGE LANGUAGE MODELS

### 5.1 EXPERIMENTAL SETUPS

**Datasets**. Our instruction tuning data are derived from Super-Natural Instructions (Wang et al., 2022), which are part of the FLAN v2 benchmark (Chung et al., 2024). We selected 17 representative tasks, comprising 9 natural language generation (NLG) tasks and 8 natural language understanding (NLU) tasks. Following the methodology of Wang et al. (2022), we randomly designated 4 datasets as held-out test sets and used the remaining 13 datasets for training. Each training dataset contains 150 examples, and each test dataset contains 100 examples, resulting in a training set of 1,950 examples and a test set of 400 examples, as summarized in Table 2.

Table 2: Summary of datasets.

| Split | Category | # Tasks | # Samples |
|---|---|---|---|
| Training | NLG | 7 | 1050 |
| | NLU | 6 | 900 |
| Testing | NLG | 2 | 200 |
| | NLU | 2 | 200 |

**Evaluation Metrics**. Following the practice in Super-Natural Instructions (Mishra et al., 2022; Wang et al., 2022), we adopt ROUGE-L (Lin, 2004) for reporting performance results, due to the diversity of our tasks and the open-ended nature of instruction tuning. We also report a single "average" metric across all datasets, following the methodology in FLAN (Wei et al., 2022; 2023).

**Baselines and Models**. We evaluate our framework against several learning algorithms: Empirical Risk Minimization (**ERM**) (Min et al., 2022), ERM with Demonstration Shuffling (**ERM+DS**) and ERM with Instance Mixup (**ERM+IM**) (Zhang et al., 2018), and **InfoAC** (Xiang et al., 2024). We implement FLAN-large as the P-Net and evaluate across five LLMs: **Llama3-8B**, **Llama2-7B/13B**, **Mistral-7B**, and **Gemma-7B**. The implementation details are provided in Appendix B.

### 5.2 EVALUATION RESULTS

We evaluate PEARL from three perspectives: (1) comparison with training-stage methods, (2) generalization to diverse type of LLMs, and (3) scalability to many-shot in-context learning (Agarwal et al., 2024) and long sequences.

Table 3: Average and Worst-Case Performance of Llama3-8B on four held-out tasks: Commonsense-QA (CSQA), Curiosity Dialogue (CurDial), CoLA, and Tell Me Why (TMW). Performance improvements (%) over ERM shown in blue. Worst-case performance tested using exhaustive search.

| # Shot | Method | Average | | CSQA | | CurDial | | CoLA | | TMW | |
|---|---|---|---|---|---|---|---|---|---|---|---|
| | | Avg. | Worst. | Avg. | Worst. | Avg. | Worst. | Avg. | Worst. | Avg. | Worst. |
| 2 | ERM | 57.3 | 49.4 | 58.0 | 54.0 | 57.9 | 43.4 | 62.0 | 58.0 | 51.1 | 42.0 |
| | ERM+DS | 57.5 (-0.2) | 48.6 (-1.6) | 62.0 | 54.0 | 54.1 | 37.8 | 61.0 | 60.0 | 51.5 | 42.7 |
| | ERM+IM | 53.5 (-6.6) | 44.4 (-10.1) | 63.0 | 54.0 | 44.7 | 28.1 | 57.0 | 56.3 | 49.4 | 39.2 |
| | INFOAC | 55.7 (-2.9) | 47.6 (-3.7) | 57.5 | 56.0 | 53.4 | 36.4 | 63.0 | 61.5 | 48.7 | 37.3 |
| | PEARL | 62.9 (+9.8) | 56.4 (+14.2) | 65.0 | 62.0 | 60.3 | 50.7 | 71.0 | 68.0 | 55.1 | 44.8 |
| 3 | ERM | 57.8 | 38.3 | 57.7 | 47.0 | 61.4 | 25.9 | 61.9 | 52.0 | 50.3 | 29.4 |
| | ERM+DS | 56.1 (-2.9) | 39.7 (+3.7) | 60.0 | 46.0 | 54.1 | 25.4 | 60.0 | 56.0 | 50.3 | 31.5 |
| | ERM+IM | 55.3 (-4.3) | 39.8 (+3.9) | 59.0 | 46.0 | 54.6 | 28.0 | 57.6 | 53.1 | 50.0 | 31.9 |
| | INFOAC | 56.3 (-2.6) | 39.5 (+3.1) | 59.3 | 49.0 | 55.2 | 24.3 | 62.1 | 55.8 | 48.4 | 28.8 |
| | PEARL | 63.1 (+9.2) | 46.9 (+22.5) | 68.4 | 62.0 | 66.7 | 34.8 | 64.7 | 56.0 | 52.4 | 34.7 |
| 4 | ERM | 59.7 | 30.6 | 61.3 | 38.0 | 62.9 | 21.3 | 63.3 | 45.8 | 51.1 | 17.5 |
| | ERM+DS | 57.7 (-3.4) | 31.8 (+3.9) | 63.3 | 40.0 | 57.3 | 17.6 | 60.1 | 52.0 | 49.9 | 17.8 |
| | ERM+IM | 56.0 (-6.2) | 32.4 (+5.9) | 63.2 | 42.0 | 53.7 | 17.8 | 57.6 | 48.5 | 49.6 | 21.3 |
| | INFOAC | 58.6 (-1.8) | 33.0 (+7.8) | 63.7 | 44.0 | 58.7 | 19.0 | 63.9 | 51.0 | 48.1 | 17.0 |
| | PEARL | 63.1 (+5.7) | 39.6 (+29.4) | 68.4 | 52.0 | 69.2 | 31.3 | 64.7 | 52.0 | 50.1 | 23.0 |

Table 3 presents the comparative performance of PEARL against other learning methods. PEARL consistently improves both average and worst-case performance across all unseen tasks. As the number of shots increases, the worst-case performance gain relative to ERM progressively increases from 14.2% at two shots to 29.4% at four shots. Notably, while optimized for worst-case performance, PEARL also achieves superior average performance with gains of 5.7% to 9.8%. This improvement may stem from the rapid convergence observed during Llama-7B's fine-tuning, where the training loss plateaus within one epoch. The rapid convergence suggests that for advanced LLMs like LLama3, focusing on challenging permutations during training is more effective than using random ones—an observation consistent with Xu et al. (2024).

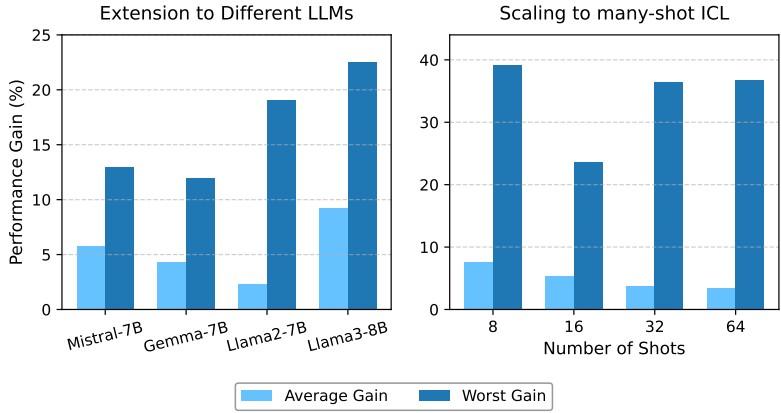

Figure 5: Generalization performance of our method across different types of LLMs and many-shot settings. **Left**: Performance gains on 3-shot across different LLMs (Mistral-7B, Gemma-7B, Llama 2-7B, and Llama3-8B). **Right**: Scaling behavior across many-shot settings (8, 16, 32, and 64 shots) and longer sequences (8k tokens) when trained with 5 shots and a sequence length of 512 tokens.

To validate the general applicability of our method, we expanded our experiments to include three additional LLMs: Mistral-7B, Gemma-7B and Llama 2-7B. As shown in the left panel of Figure 5, our method consistently improves worst-case performance by more than 10% in three-shot settings. Additional results for higher-shot settings are provided in Appendix D. We also observed that different LLM families exhibit varying sensitivity to input permutations, with Llama models being the most sensitive, followed by Gemma and Mistral. Despite these differences, the phenomenon remains significant, with performance drops exceeding 10% in most cases. Notably, our method achieves consistent worst-case performance improvements of over 10% for three or more shots, demonstrating its robustness across diverse model families.

We scale our evaluation to the many-shot ICL setting (up to 64 shots) and longer sequences (up to 8,000 tokens) after training with 5 shots and a sequence length of 512 tokens. As shown in the right panel of Figure 5, PEARL achieves substantial worst-case performance gains ranging from 24% to 40% when generalizing to larger shot numbers and longer sequences, despite being trained on a smaller setup. These results suggest that PEARL enables LLMs to learn robust features that generalize effectively to both many-shot in-context learning and longer sequences, demonstrating the strong generalization capability of our method. Detailed results are provided in Appendix E.

Table 4: Shot Efficiency: Average Performance with and without PEARL.

| # Shots | 2 | 4 | 8 | 16 | 32 | 64 |
|---|---|---|---|---|---|---|
| ERM | 57.3 | 59.7 | 61.8 | 66.9 | 67.4 | 68.1 |
| PEARL | 62.9 | 63.1 | 66.5 | 70.5 | 70.0 | 70.4 |

As we scale to the many-shot setting, we also observe notable trends in *shot efficiency*, which quantifies the number of shots a baseline model would require to match the average performance of a PEARL-trained model. As shown in Table 4, PEARL-trained models achieve comparable average performance while requiring *two to four times fewer shots*, highlighting the efficiency of our approach.

## 6 RELATED WORK

**Order Sensitivity in In-context Learning** Despite the huge success of ICL, its robustness to demonstration permutations remains an unresolved challenge (Zhao et al., 2021).

Most *training-stage methods* focus on improving general performance in ICL (Min et al., 2022; Wei et al., 2023) while neglecting the lack of robustness to the permutations of demonstrations. Recent studies suggest that this phenomenon stems from the autoregressive nature of transformer language models (Chen et al., 2023b; Xiang et al., 2024). InfoAC (Xiang et al., 2024) introduces contrastive learning during fine-tuning to break the autoregressive constraint and enable bidirectional token visibility; however, their approach achieves limited success and is restricted to classification tasks. Preliminary work of Chen et al. (2023b) shows the DeepSet architecture exhibits better permutation invariance than transformer; however, this MLP-based new architecture is too small to solve complex language modeling tasks. *Our approach falls within the category of training-stage methods but proposes a general learning framework that enhances permutation robustness in LLMs without modifying the Transformer architecture or its autoregressive objective, thereby preserving scalability.*

*Inference-stage methods* can be categorized into four types: (1) demonstration selection (Chang & Jia, 2023; Peng et al., 2024), which improves normal-case performance but lacks worst-case guarantees under permutations; (2) output calibration (Zhao et al., 2021; Li et al., 2023; Guo et al., 2024a), which are effective for classification but is less applicable to generation tasks due to sequence calibration challenges; (3) order optimization (Lu et al., 2022), which finds the best ordering during inference but suffers from exponential complexity; and (4) prediction ensembling: A recent work (Zhang et al., 2024) transforms n-shot ICL into n one-shot predictions and ensembles results—effective for classification but harms generation. In summary, inference-stage methods mitigate order sensitivity via pre/post-processing, often introducing additional inference overhead. Moreover, most methods target classification and underperform on generation tasks. *In contrast, our training-stage solution complements inference-stage methods, enhancing LLM robustness without additional inference costs while remaining broadly applicable to various tasks.*

**Distributionally Robust Optimization**. Distributionally robust optimization optimizes the objective function over ambiguity sets, often defined as balls centered on the empirical distribution (Ben-Tal et al., 2013; Lam & Zhou, 2015; Duchi et al., 2016; Miyato et al., 2018). Prior applications of DRO have primarily addressed distributional shifts, including label shift (Hu et al., 2018) and data source shift (Oren et al., 2019) and group shift (Sagawa et al., 2020). To the best of our knowledge, we are the first to apply DRO to enhance the ICL robustness of LLMs by defining the ambiguity set over all possible permutations of the empirical distribution, thereby providing performance guarantees.

**Optimal Transport**. Optimal transport is a fundamental mathematical discipline established by Monge (1781); Kantorovich (1942). It defines a metric for measuring distances between probability distributions, known as the Wasserstein distance, and has been widely employed in machine learning for distribution matching (Montesuma et al., 2024; Xiao et al., 2024). Our work extends the concept of learning permutation structures through neural networks, as explored in (Mena et al., 2018) for learning to sort numbers or solve jigsaw puzzles. However, we apply OT in the context of LLMs and design a neural network, P-Net, equipped with the Sinkhorn operator to generate challenging permutations, enabling LLMs to undergo DRO training and thereby enhancing their ICL robustness.

## 7 CONCLUSION

We introduce a novel permutation-resilient learning framework, PEARL, designed to enhance the robustness of LLMs against different permutations. PEARL employs a permutation-proposal network, which leverages the Sinkhorn algorithm to generate challenging permutations, optimized under the DRO to systematically improve LLM's robustness. Through empirical evaluations on both synthetic tasks and real-world instruction-tuning tasks, our framework has demonstrated effectiveness in mitigating the permutation-based attacks and enhancing average performance.

While PEARL primarily focuses on improving in-context learning, it provides a general framework for handling set-structured inputs with order-independent elements, such as multiple documents, images, or videos. We hope this work inspires further research on permutation-resilient learning, contributing to the development of more robust and trustworthy language models.

ACKNOWLEDGMENTS

We sincerely thank the anonymous reviewers for their valuable feedback and constructive suggestions, which have helped improve the quality of this work. This work is partially supported by Hong Kong RGC GRF No. 14206324, CUHK direct grant No. 4055209, and CUHK Knowledge Transfer Project Fund No. KPF23GWP20. Li Shen is supported by Shenzhen Basic Research Project (Natural Science Foundation) Basic Research Key Project (NO. JCYJ20241202124430041), and CCF-DiDi GAIA Collaborative Research Funds.

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

# APPENDIX

## A DETAILED SETUP OF ICL WITH LINEAR FUNCTIONS

### A.1 DATASETS CONSTRUCTION

We investigate training a language model to perform in-context learning on linear functions, following Garg et al. (2022); Guo et al. (2024b). The function class is defined as $\mathcal{F} = \{f \mid f(x) = w^\top x, w \in \mathbb{R}^d\}$, where $d$ is the input dimension. Each data sample is constructed as follows:

(a) Function sampling: A weight vector $w \sim \mathcal{N}(0, I_d)$ is sampled, defining a linear function $f(x) = w^\top x$.

(b) Input sampling: Inputs $x_1, x_2, \ldots, x_{k+1} \sim \mathcal{N}(0, I_d)$ are independently drawn.

(c) Output generation: For each input, the corresponding output is computed as $y_i = f(x_i) = w^\top x_i$ for $i = 1, 2, \ldots, k + 1$.

The input prompt $p^i$ consists of $i$ demonstrations and the $(i + 1)$-th example as the query: $p^i = (x_1, f(x_1), x_2, f(x_2), ..., x_i, f(x_i), x_{i+1})$. We trained a language model $L_\theta$, parameterized by $\theta$, to minimize the expected loss over all input prompts:

$$\min_\theta \mathbb{E}_p \left[ \frac{1}{k+1} \sum\nolimits_{i=0}^k \ell(\theta; p^i, f(x_{i+1})) \right], \tag{18}$$

where $l(\cdot)$ is the mean squared error (MSE) loss. During testing, we evaluated performance using the same MSE metric. We report the normalized squared error $((LM(p) - w^\top x_{\text{query}})^2/d)$, where $d$ is the problem dimension.

### A.2 IMPLEMENT DETAILS

**Architecture**. Following Garg et al. (2022), we implement $L_\theta$ using a GPT-2 architecture (Radford et al., 2019) with 12 layers, 8 attention heads, and a hidden dimension of 256. The model takes as input a sequence of vectors in its embedding space and predicts the next vector in the sequence within the same space.

**Training**. We pre-train the model from scratch on a generated dataset of 40k linear functions using the AdamW (Loshchilov & Hutter, 2019). We employ a batch size of 128 and trained for 500k steps, selecting the best checkpoint based on validation set performance. In the PEARL framework, we randomly initialize the P-Net with a BERT-base-sized transformer encoder, also pre-training it from scratch. During testing, we sample novel functions to assess the model's ability to infer new weights $w$ through in-context demonstrations.

## B DETAILED SETUP OF INSTRUCTION FINE-TUNING

### B.1 DETAILS OF DATASETS

The details of datasets used in instruction tuning is presented in Table 5.

### B.2 BASELINE AND IMPLEMENTATION DETAILS

To evaluate the performance of our trained model, we compare it with other learning algorithms.

**Empirical Risk Minimization (ERM)** (Min et al., 2022): Standard approach minimizing the average loss over the training dataset, adopted by mainstream instruction tuning models such as FLAN (Chung et al., 2024), Natural Instructions (Mishra et al., 2022; Wang et al., 2022), and MetaICL (Min et al., 2022).

**ERM with Demonstration Shuffling (ERM+DS)** (Zhang et al., 2018): Enhances ERM by randomly shuffling the order of in-context demonstrations within each sample at each training step. This introduces robustness by exposing the model to different permutations of demonstrations during training. It can be considered a form of epoch-level data augmentation.

Table 5: Details of datasets used in instruction tuning from natural instructions.

| Task ID | Task Name | Source | Category |
|---|---|---|---|
| 1297 | QASC Question Answering | QASC | Question Answering |
| 442 | COM_QA Paraphrase Question Generation | COM_QA | Question Rewriting |
| 908 | DialogRE Identify Familial Relationships | DialogRE | Speaker Relation Classification |
| 288 | Gigaword Summarization | Gigaword | Title Generation |
| 582 | Natural Questions Answer Generation | Natural Questions | Question Answering |
| 151 | TOMQA Find Location Easy Clean | TOM_QA | Question Answering |
| 1714 | ConvAI3 Sentence Generation | ClariQ | Dialogue Generation |
| 379 | AGNews Topic Classification | AG News | Text Categorization |
| 639 | MultiWOZ User Utterance Generation | MultiWOZ 2.2 | Dialogue Generation |
| 209 | Stance Detection Classification | StarCon | Stance Detection |
| 1516 | IMPPRES Natural Language Inference | IMPPRES | Textual Entailment |
| 589 | Amazon Food Summary Text Generation | Amazon Reviews | Summarization |
| 1285 | KPA Keypoint Matching | ArgKP | Text Matching |

**ERM with Instance Mixup (ERM+IM)**(Zhang et al., 2018): Incorporates *Instance Mixup* technique during each training step. For each data point, we generate multiple augmented versions by randomly selecting different in-context demonstrations. We perform multiple forward passes to compute the loss for each augmented version, average these losses, and then perform a single backward pass using the averaged loss. This approach provides finer-grained data augmentation compared to demonstration shuffling. Notably, by comparing this baseline with our method, we contrast min-mean optimization (ERM+IM) with min-max optimization (our method).

**InfoAC**: (Xiang et al., 2024) is a training-stage method that employs contrastive learning to enable earlier tokens to access information from later tokens, amining to mitigate the order sensitivity of ICL inherent in autoregressive LM.

By including these baselines we provide a comprehensive evaluation of our proposed method.

As for the proposed PEARL framework, we select the LLaMA3-8B model as our LLM and the FLAN-large encoder as the P-Net. Both models are fine-tuned using LoRA (Hu et al., 2022), with the number of finetuned parameters of P-Net being 1/20 that of the LLM. We train the models on the instruction dataset for two epochs using a single NVIDIA A40 GPU, with a batch size of 16, resulting in a total of 246 training steps. The optimizer used was AdamW. The learning rates for the P-Net and the LLM are set to $1 \times 10^{-4}$ and $3 \times 10^{-4}$, respectively. For the Sinkhorn algorithm, we use 80 iterations, a temperature parameter of 0.1, and an entropy constraint coefficient $\beta = 1.0$.

### B.3 DETAILS OF HYPERPARAMETER SETTINGS

In this section, we provide a comprehensive overview of the hyperparameter settings used in our experiments (Table 6). The hyperparameters can be categorized into three groups: (1) basic LLM training parameters, such as learning rate and batch size; (2) LoRA configuration parameters; and (3) P-Net optimization parameters. These hyperparameters were selected based on average validation performance and kept consistent across comparative experiments to ensure fair comparison.

## C ANALYSIS OF HYPERPARAMETERS IN INSTRUCTION FINETUNING

We conduct analysis to understand the impact of key hyperparameters on P-Net learning and our overall framework. Our analysis focuses on two main aspects: the effect of the entropy constraint strength, and the influence of iteration number and temperature in the Sinkhorn algorithm.

**Influence of Entropy Regularization in OT** We examine the impact of the entropy regularization coefficient in OT, testing values of 0.3, 1.0, 3.0, and 10.0 (Figure 7). At a low coefficient (0.3), P-Net's gradient norm remained small, indicating minimal learning and potential generation of simple semantic overlaps to satisfy adversarial training requirements. Concurrently, the LLM's gradient norm struggled to decrease. The gradient norm for P-Net peaked at 1.0, suggesting optimal learning conditions. As coefficients increased to 3.0 and 10.0, P-Net's gradient norm decreased again, suggesting excessive restrictions. The range of 1.0-3.0 provided an ideal balance, encouraging

| Category | Hyperparameter | Value |
|---|---|---|
| LLMs | Learning rate | 3e-5 |
| | Batch size | 16 |
| | Max sequence length | 512 |
| | Weight decay coefficient | 0.1 |
| | Epoch | 2 |
| LoRA | Rank | 8 |
| | Alpha | 32 |
| | Dropout | 0.1 |
| | P-Net target modules | q, v |
| | LLMs target modules | q_proj, k_proj, v_proj, o_proj, gate_proj, up_proj, down_proj |
| P-Net | Temperature | 0.1 |
| | Iteration coefficient | 80 |
| | Entropy constraint | 1.0 |
| | Noise | 0.3 |
| | Learning rate | 1e-4 |
| | Batch size | 16 |
| | Max sequence length | 512 |

Table 6: Hyperparameter settings used in our main experiment.

Figure 6: Impact of number of iterations and temperature on the average/worst-case performance.

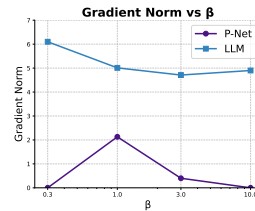

| # Iter. | Temperature | | |
|---|---|---|---|
| | 0.03 | 0.1 | 0.3 |
| 80 | 55.7 / 40.0 | 55.7 / 40.0 | 55.4 / 39.6 |
| 200 | 55.7 / 40.0 | 55.8 / 40.0 | 55.8 / 40.6 |

Figure 7: Impact of entropy coefficient.

P-Net to extract meaningful information from the LLM without oversimplifying or overcomplicating the task. In contrast, the LLM's gradient norm decreased consistently with increasing coefficients, indicating a distinct response to entropy regularization.

**Effect of Sinkhorn Algorithm Parameters** We investigate the interplay between two critical parameters in the Sinkhorn algorithm: number of iterations and temperature. Intuitively, these parameters are positively correlated; higher iteration counts typically allow for higher temperatures. Our experiments, however, reveal an unexpected robustness to parameter variations. With the entropy regularization coefficient fixed at 1, we vary the number of iterations $(80, 200)$ and temperature $(0.03, 0.1, 0.3)$. As presented in Table 6, surprisingly, these substantial parameter changes result in minimal performance variation. This suggests that the Sinkhorn algorithm in our framework is less sensitive to these parameters than initially hypothesized, potentially indicating a wider range of stable configurations for practical applications.

# D    EXTENDED INSTRUCTION FINETUNING ACROSS DIVERSE LLMS

We expanded our evaluation to more LLMs: Mistral-7B, Gemma-7B, and earlier generations such as Llama2-7B and Llama2-13B, as detailed in the tables from Table (7) to Table (10).

**Sensitivity to Permutations Across LLM Families**    Our analysis reveals that different LLM families exhibit varying degrees of sensitivity to permutations. The sensitivity ranking, from highest to lowest, is as follows: Llama, Gemma, and Mistral. Notably, all examined families showed significant performance declines, typically exceeding 10 %.

Table 7: Instruction fine-tuning results for Mistral-7B evaluated on four held-out tasks. Performance gains (%) over the ERM baseline are indicated in blue.

| # Shot | Method | Average | | CSQA | | CurDial | | CoLA | | TMW | |
|---|---|---|---|---|---|---|---|---|---|---|---|
| | | Avg. | Worst. | Avg. | Worst. | Avg. | Worst. | Avg. | Worst. | Avg. | Worst. |
| 2 | ERM | 64.1 | 58.1 | 67.0 | 64.0 | 54.6 | 41.8 | 81.0 | 78.0 | 53.7 | 48.5 |
| | PEARL | 67.0 (+4.5) | 62.4 (+7.5) | 68.0 | 66.0 | 59.4 | 49.0 | 82.0 | 78.0 | 58.4 | 56.7 |
| 3 | ERM | 66.6 | 56.1 | 67.0 | 62.0 | 63.7 | 38.9 | 80.0 | 76.0 | 55.6 | 47.3 |
| | PEARL | 69.5 (+4.3) | 62.8 (+12.0) | 70.0 | 66.0 | 70.1 | 60.1 | 83.6 | 78.0 | 54.1 | 47.0 |
| 4 | ERM | 66.7 | 50.4 | 68.9 | 60.0 | 67.6 | 47.8 | 74.2 | 52.0 | 55.9 | 41.6 |
| | PEARL | 68.3 (+2.5) | 57.1 (+13.4) | 69.9 | 62.0 | 71.6 | 54.8 | 74.9 | 66.0 | 56.8 | 45.5 |
| 5 | ERM | 67.9 | 50.7 | 67.5 | 56.0 | 70.7 | 52.6 | 76.0 | 56.0 | 57.4 | 38.2 |
| | PEARL | 70.2 (+3.4) | 58.1 (+14.5) | 70.4 | 64.0 | 76.7 | 59.3 | 73.3 | 66.0 | 60.4 | 43.0 |

Table 8: Instruction fine-tuning results for Gemma-7B evaluated on four held-out tasks. Performance gains (%) over the ERM baseline are indicated in blue.

| # Shot | Method | Average | | CSQA | | CurDial | | CoLA | | TMW | |
|---|---|---|---|---|---|---|---|---|---|---|---|
| | | Avg. | Worst. | Avg. | Worst. | Avg. | Worst. | Avg. | Worst. | Avg. | Worst. |
| 2 | ERM | 66.2 | 59.5 | 71.0 | 70.0 | 59.1 | 46.1 | 77.0 | 70.0 | 57.8 | 52.0 |
| | PEARL | 66.3 (+0.0) | 60.7 (+2.0) | 74.0 | 68.0 | 47.3 | 39.2 | 82.0 | 78.0 | 61.7 | 57.6 |
| 3 | ERM | 64.7 | 52.5 | 70.7 | 64.0 | 67.1 | 45.2 | 70.3 | 60.0 | 50.5 | 40.7 |
| | PEARL | 68.4 (+5.8) | 59.3 (+13.0) | 74.7 | 68.0 | 59.2 | 42.5 | 78.7 | 76.0 | 61.0 | 50.6 |
| 4 | ERM | 65.0 | 46.5 | 65.0 | 54.0 | 71.4 | 41.1 | 72.5 | 58.0 | 51.1 | 32.9 |
| | PEARL | 67.2 (+3.4) | 52.5 (+13.0) | 71.4 | 60.0 | 60.7 | 38.9 | 75.9 | 66.0 | 60.8 | 45.2 |
| 5 | ERM | 64.3 | 46.3 | 65.9 | 54.0 | 73.4 | 48.3 | 65.6 | 50.0 | 52.3 | 32.9 |
| | PEARL | 66.3 (+3.1) | 51.0 (+10.2) | 70.3 | 60.0 | 63.4 | 43.6 | 71.3 | 60.0 | 60.2 | 40.4 |

Table 9: Instruction fine-tuning results for Llama2-7B evaluated on four held-out tasks. Performance gains (%) over the ERM baseline are indicated in blue.

| # Shot | Method | Average | | CSQA | | CurDial | | CoLA | | TMW | |
|---|---|---|---|---|---|---|---|---|---|---|---|
| | | Avg. | Worst. | Avg. | Worst. | Avg. | Worst. | Avg. | Worst. | Avg. | Worst. |
| 2 | ERM | 56.6 | 46.3 | 56.0 | 50.0 | 61.3 | 50.2 | 58.2 | 42.0 | 50.7 | 43.1 |
| | PEARL | 57.4 (+1.5) | 46.5 (+0.4) | 58.0 | 48.0 | 55.2 | 44.7 | 62.0 | 48.0 | 54.4 | 45.4 |
| 3 | ERM | 58.2 | 34.0 | 52.7 | 34.0 | 64.0 | 36.4 | 66.0 | 36.0 | 50.1 | 29.4 |
| | PEARL | 59.6 (+2.3) | 40.4 (+19.1) | 56.3 | 40.0 | 66.2 | 46.2 | 67.0 | 42.0 | 48.7 | 33.5 |
| 4 | ERM | 58.9 | 19.9 | 60.0 | 26.0 | 68.1 | 24.4 | 60.2 | 14.0 | 47.3 | 15.1 |
| | PEARL | 60.5 (+2.7) | 31.6 (+59.1) | 61.2 | 40.0 | 69.4 | 40.1 | 62.4 | 24.0 | 48.9 | 22.4 |
| 5 | ERM | 61.9 | 25.8 | 59.0 | 32.0 | 74.2 | 43.9 | 65.7 | 10.0 | 48.6 | 17.1 |
| | PEARL | 62.9 (+1.6) | 32.1 (+24.7) | 62.4 | 38.0 | 73.3 | 43.4 | 64.8 | 24.0 | 51.0 | 23.0 |

Table 10: Instruction fine-tuning results for Llama2-13B evaluated on four held-out tasks. Performance gains (%) over the ERM baseline are indicated in blue.

| # Shot | Method | Average | | CSQA | | CurDial | | CoLA | | TMW | |
|---|---|---|---|---|---|---|---|---|---|---|---|
| | | Avg. | Worst. | Avg. | Worst. | Avg. | Worst. | Avg. | Worst. | Avg. | Worst. |
| 2.0 | ERM | 66.3 | 56.6 | 56.0 | 46.0 | 72.6 | 56.2 | 83.0 | 76.0 | 53.4 | 48.0 |
| | PEARL | 67.9 (+2.4) | 60.7 (+7.3) | 64.0 | 58.0 | 73.8 | 64.2 | 81.0 | 76.0 | 52.6 | 44.4 |
| 3.0 | ERM | 65.7 | 46.2 | 55.7 | 38.0 | 76.4 | 51.3 | 77.7 | 56.0 | 53.1 | 39.6 |
| | PEARL | 68.5 (+4.2) | 50.3 (+8.7) | 62.7 | 44.0 | 81.0 | 58.4 | 76.7 | 56.0 | 53.5 | 42.6 |
| 4.0 | ERM | 65.8 | 33.2 | 58.2 | 28.0 | 79.6 | 41.6 | 73.7 | 38.0 | 51.8 | 25.0 |
| | PEARL | 66.4 (+0.9) | 40.2 (+21.1) | 63.3 | 42.0 | 80.4 | 45.5 | 69.4 | 42.0 | 53.1 | 29.1 |

**Adaptation of the Proposed Method**   In scenarios with three or more examples, our method consistently demonstrated substantial improvements, often enhancing worst-case performance by more than 10%. These results confirm the robustness and effectiveness of our approach.

## E   SCALING TO MANY-SHOT IN-CONTEXT LEARNING

We evaluate the scalability of PEARL by extending our analysis to many-shot scenarios, testing performance with 8 to 64 in-context examples (Table 11). Notably, despite being trained solely on 5-shot demonstrations, PEARL exhibits strong generalization to settings with substantially more examples. Using Llama3-8B as our base model, we compare PEARL and ERM training approaches across four held-out tasks. Our analysis reveals persistent performance advantages of PEARL over the ERM baseline across all shot regimes.

Table 11: Performance evaluation across 8-, 16-, 32-, and 64-shot settings comparing PEARL and ERM learning algorithm for Llama3-8B on four held-out tasks, with gains (%) relative to the ERM.

| # Shot | Method | Average | | CSQA | | CurDial | | CoLA | | TMW | |
|---|---|---|---|---|---|---|---|---|---|---|---|
| | | Avg. | Worst. | Avg. | Worst. | Avg. | Worst. | Avg. | Worst. | Avg. | Worst. |
| 8 | ERM | 61.8 | 21.3 | 61.4 | 36.0 | 68.3 | 22.7 | 62.7 | 16.0 | 54.8 | 10.6 |
| | PEARL | 66.5 (+7.6) | 29.7 (+39.2) | 67.7 | 44.0 | 77.1 | 28.7 | 65.0 | 32.0 | 56.2 | 14.0 |
| 16 | ERM | 66.9 | 21.3 | 67.3 | 36.0 | 76.5 | 31.4 | 67.2 | 8.0 | 56.5 | 9.7 |
| | PEARL | 70.5 (+5.3) | 26.3 (+23.7) | 70.9 | 46.0 | 83.9 | 37.5 | 70.1 | 12.0 | 56.9 | 9.8 |
| 32 | ERM | 67.4 | 19.3 | 67.5 | 32.0 | 77.8 | 30.7 | 68.2 | 6.0 | 56.1 | 8.6 |
| | PEARL | 70.0 (+3.8) | 26.4 (+36.4) | 70.0 | 44.0 | 82.6 | 40.3 | 70.6 | 12.0 | 56.6 | 9.1 |
| 64 | ERM | 68.1 | 20.6 | 68.1 | 38.0 | 76.9 | 27.7 | 72.2 | 8.7 | 55.0 | 8.0 |
| | PEARL | 70.4 (+3.5) | 28.2 (+36.7) | 69.5 | 46.0 | 82.9 | 38.9 | 74.2 | 19.6 | 55.1 | 8.1 |

## F   BEST-CASE PERFORMANCE

Although our methodology was initially designed to optimize for pessimistic (worst-case) scenarios, we have also included an evaluation of the best-case performance for both PEARL and ERM to provide a balanced perspective. The results are shown in the Table 12.

Table 12: Best performance comparison between ERM and PEARL

| #Shot | Method | Average | Gain | CSQA | CurDial | CoLA | TMW |
|---|---|---|---|---|---|---|---|
| 2 | ERM | 64.1 | - | 68.8 | 64.4 | 64.1 | 59.2 |
| | PEARL | 68.8 | 7.2% | 73.4 | 69.2 | 70.3 | 62.1 |
| 3 | ERM | 72.8 | - | 70.3 | 85.0 | 65.6 | 70.3 |
| | PEARL | 77.0 | 5.7% | 73.4 | 87.9 | 79.7 | 66.9 |
| 4 | ERM | 82.9 | - | 81.3 | 92.4 | 78.1 | 79.7 |
| | PEARL | 84.3 | 1.7% | 82.8 | 93.6 | 81.2 | 79.5 |
| 5 | ERM | 86.8 | - | 84.4 | 95.3 | 81.3 | 86.2 |
| | PEARL | 89.3 | 2.9% | 87.5 | 96.5 | 85.9 | 87.3 |

Surprisingly, the results show that across all datasets and in every shot condition, PEARL's best performance consistently exceeded that of ERM. This indicates that our method not only optimizes performance in worst-case scenarios but also slightly enhances best-case performance.

