# OpenReview forum: "PEARL: Towards Permutation-Resilient LLMs"
_ICLR.cc/2025/Conference — ICLR 2025 Poster_

### Official Review · Reviewer_n5jf · 2024-10-28

**Soundness:** 2
**Presentation:** 3
**Contribution:** 2
**Rating:** 3
**Confidence:** 4

**Summary:**

This paper considers the tendency for the performance of LLMs to decrease given changes to the ordering of in-context (IC) examples. The authors propose an approach based on distributionally robust optimization (DRO), which they then run on both synthetically generated data as well as standard benchmarks. They show that performance against adversarial modifications of the ordering of the IC examples tends to increase when fine-tuning using their DRO approach.

**Strengths:**

* The paper has a relatively strong set of experiments. If you agree that the threat model is worth studying, then you would certainly conclude that the method provides clear robustness against this kind of attack.
* Aside from a few minor typos, the paper is well-written and easy to follow.

**Weaknesses:**

**Threat model**

* "This paper shows that this vulnerability can be exploited to design a natural and completely imperceptible attack. . ." -- This claim seems untrue. The attack is perceptible; the attack changes the order of the IC examples, and a keen user could very well notice this. Or, if the argument is that the user never sees the IC examples, then of course it's imperceptible. But *any* attack that is hidden from the user by construction would be imperceptible, so this doesn't seem like a distinguishing aspect of the algorithm.
* "Consequently, there remains a significant need for methods that can fundamentally enhance LLMs’ inherent ability to manage the vast combinatorial space of possible input permutations." -- I'm not sure I'm convinced. Why is the need so pronounced, and in what realistic setting do you imagine that an adversary will have access to the IC examples? It seems quite unlikely that an adversary would have access to the same context window as a user. Moreover, even if the IC examples go in the system prompt, it's unclear in what settings an adversary would have this kind of access, particularly if one tends to use proprietary models, which tend not to expose their system prompts to users. Indeed, this accounts for the vast majority of use cases, so calling the need "significant" seems too strong. My advice: The authors should motivate their threat model and clearly explain why it corresponds to a real-world setting. Perhaps a demonstration of this attack happening in the wild, especially in the first few sections of this paper, would make sense.

**Experiments.**
* "We select two tasks. . . We test 100 samples from each task." -- Is 100 samples sufficiently many? I'd be concerned that you may not get stable results (i.e., the variance would be relatively high) on datasets this small. Did you run multiple independent trials? How were these 100 examples selected?
* It would be worth using more than one LLM, especially in the experiments described in Figure 3. The authors use Llama 3.1, which is indeed one of the strongest one-source models. It seems that one could also simulate this attack on proprietary models by putting these examples in the context window before the user's prompt. Is this the case, or have I misunderstood something?  I understand that the defense proposed later on requires fine-tuning the model, but even here, it would be worth extending the score of the analysis to include other open-source models to get a better sense for how this vulnerability impacts models outside of the Llama family.
* The drop in performance seems to be a factor not only of $\tau$, but also of the number of examples, since (as the authors point out), larger $n$ would tend to mean that the lower bound on performance is lower.
* In eq. (1), how are $\mu_i$ and $w_i$ calculated? Is this meant to be a general overview of the method, or something more specific to the experiments in Figure 1. Indeed, it may also be worth characterizing $R$ as a function of $D$ and $\tau$, i.e., $R = R(D, \tau)$.
* I didn't understand Figure 2. In the ERM problem in eq. (2), the optimization  $\theta$ is learned, but the text indicates that both ERM and DRO learn "probability distributions of models." This doesn't make sense to me, because neither (2) nor (3) deal with a distribution over models. For this reason, the plots in Figure 2 are hard to interpret; what is $\hat{P}$ (as far as I know, this isn't defined in the text), is this different from $\hat{p}$ (referred to in the caption), and what is $P_\theta$? It would be worth explaining the example more as well. It's unclear why only indices 2,3, and 5 are considered here.  Moreover, the bars in both plots also look exactly the same, which I'd assume isn't the right message to take away.
* The paper would benefit from a clearer explanation regarding how the hyperparameters are chosen.
* It would be worth describing how the Rogue-L metric is calculated and why it's a good metric to use here.

**Proposed method.**
* It'd be worth saying more about why this instantiation of the problem is NP-Hard. Is it simply because I would need to enumerate all of the options in $\mathcal{Q}$ (assuming that's what $\mathcal{Q}$ is, since it's rigorously defined in the text)?
* The [CLS] token seems to appear from nowhere. What does CLS stand for?
* It would be worth describing more why the proposed approach corresponds to *adversarial* training. The approach seems to involve minimizing a certain loss function, rather than the typical zero-sum formulation of adversarial training which is standard in the computer vision/adversarial examples literature.
* The problem is a tad bit underspecified. The authors should be precise about what the dimensionality of $X$ is and what $N$ is in the formulation on page 6.  It's also unclear why the authors want to add randomness (c.f., "To introduce randomness and control..."); why is randomness needed in this algorithm, and why, in particular, is the current method the right way to introduce it? Also, why do we need to "control the discreteness?" I am having a hard time understanding what this means and/or why the authors are trying to do this.
* The authors say that their formulation allows the user to transform "the input permutation distribution into any permutation distribution." Again, the paper should be more clear about whether we are dealing with distributions over perturbations or perturbations themselves. You'd imagine that there is a primal view of (3) where we deal over the objects themselves, rather than distributions over these objects.
* In (8), it's unclear where the sinkhorn operator factors in. And how does this formulation connect to the DRO formulation in (3). From the perspective of the reader, this problem in (8) seems to appear out of thin air, and I'd imagine that readers will get stuck at this point.
* Unless I missed something, the variable $G$ is undefined in Algorithm 1. Is that the P-Net?
* "To prevent the P_Net from exploiting trivial solutions--such as outputting even matrices that average out the semantic content..." -- I'm not sure what this means. What is semantic content here? Isn't the P-Net outputting a permutation?
* I didn't understand the steps between 9 and 10. Consider that if we plug in the losses to

$$\min_{\theta,\phi} L(\theta){\text{LM}} + L(\phi)_{P-Net}$$

then objective becomes:

$$ \mathbb{E}[\ell(\theta; \phi; (\Pi \cdot p, x, y)) - \ell(\theta, \phi; (\pi\cdot p, x, y)) + \beta L(\theta, \phi)_\text{ent}] $$

which is equal to

$$ \mathbb{E}[ \beta L(\theta, \phi)_\text{ent}] $$

since the first two terms cancel.  Could the authors clarify this, because it seems like the loss reduces to an entropy penalty.

**Discussion of related work.**
* Outside of the related work section, there is very little discussion of related work. It would be worth justifying claims using citations, e.g., "Under appropriate assumptions, learning theory guarantees that models trained via ERM perform well on the test distribution given sufficient training data." Since this wasn't studied in the paper under review, it would be worth citing learning theoretic papers, which, if the reader had inclination, would enable looking into this matter more closely.
* It would also be worth citing representative works from papers that seek to generalize to data that is outside of the training distribution. A particularly related piece of related work is the GroupDRO paper (https://openreview.net/pdf?id=ryxGuJrFvS), which uses a similar approach in classification to the approach outlined by the authors here.

**Miscellaneous.**
* "Most existing studies on ICL primarily aim to enhance the general performance on few-shot learning." -- The paper would benefit from more precise language. "General performance" is fairly vague, and it's a bit unclear how one would distinguish this from the setting considered by the authors. Does this paper not also consider the "general performance" of ICL?
* Figure 1 -- the colors are too difficult to tell apart. It would be worth considering colors outside of the blue family.

**Questions:**

* The authors consider average and worst-case orderings of IC examples. But what about best-case ordering? This also seems like a reasonable baseline. Given that performance significantly drops between the average and worst orderings, should we expect performance to increase given a best ordering?  This almost seems like a more realistic model of LLM use than the adversarial setting: A user wants to maximize their performance, and one way of doing that is by finding a good ordering of IC examples.
* Another missing (easy) baseline is the impact of picking a *random* permutation in Figure 1. Or, is this how the average performance was calculated?
* Line 171 -- is the upper bound "simulated," or is this the actual upper bound? My understanding is that if you were to enumerate all of the permutations of the IC examples, you would be able to calculate the *true* upper bound.

---

> ### Author Response · Authors · 2024-11-23
> **Rebuttal by Authors [1/7]**
>
> Thank you for your thoughtful review and constructive feedback on our work. We appreciate your recognition of the strength of our experimental setup and are pleased that you found our paper well-written and easy to follow. Your acknowledgment that our method provides clear robustness against the ordering modifications of in-context examples is encouraging.
>
> Your insights are valuable to us, and we have made extensive revisions to the manuscript based on your suggestions. For your convenience, we have highlighted these changes in blue.
>
> We will address your comments and concerns in detail below and and have supplemented our work with new experimental results.
>
> # Response to Comment on the Threat Model
>
> Thank you for your comment and for highlighting areas that required clarification. We apologize for any misunderstanding caused by our wording, and we appreciate the opportunity to clarify the threat model.
>
> ## Clarification of the Attack Setting
> Our study focuses on a **two-party adversarial scenario** involving a malicious end-user and the model provider. In this setting, the malicious user interacts directly with the LLM and manipulates the input—for example, by permuting the order of ICL examples—to compromise the model's performance. This threat model aligns with established adversarial settings in the literature [1,2,3,4] and provides a meaningful framework for evaluating the robustness of LLMs.
>
> This differs from a **three-party attack scenario**, where a malicious actor intercepts a benign user's request. While such scenarios are also interesting, our work mainly focuses on the two-party setting where the adversary is the end-user directly interacting with the model.
>
>
> ## Clarification about "imperceptible"
>
> Thank you for bringing up this point. To clarify, the term "imperceptible" in our paper refers to being imperceptible to **model providers**, rather than end-users.
>
> In the two-party adversarial setting, a malicious user intends to attack an LLM. In this context, the malicious user prefers the attack to be less noticeable to the model provider. Most methods in this scenario involve optimized adversarial prefixes or suffixes (often nonsensical strings) that are explicitly prepended or appended to the prompt [1,2,3]
> , or introduce noise perturbations into the samples [4]. Such alterations are easily detectable and filterable by model providers.
>
> In contrast, our method involves only permuting the order of demonstrations, which is a normal operation during in-context learning and does not introduce any abnormal or suspicious content. This makes the attack less noticeable to model providers.
>
> We have updated the manuscript to include this explanation, and we hope this clarification addresses your concern.
>
> ## Response to Practical Use Scenarios
> Thank you for your feedback. We discuss the practical use scenarios of our work as follows:
>
> Our methodology is designed to enhance the robustness of LLMs in handling various possible input permutations, aiming to benefit both normal and adversarial scenarios:
>
> -   **Normal Scenario**: Typically, benign users provide demonstrations in arbitrary sequences. Our approach improves performance on the worst-case input permutations, thereby protecting the user experience in typical use cases.
>
> -   **Adversarial Scenarios**:  Malicious end-users may deliberately reorder demonstrations to exploit model vulnerabilities and induce errors. Our enhancements mitigate the risks associated with such targeted attacks by ensuring the model remains robust to permutations intended to degrade its performance.
>
> We hope these can address your concerns and demonstrates the applicability of our work.
>
> **References**
>
> [1] Universal and Transferable Adversarial Attacks on Aligned Language Models. 2023
>
> [2] Tricking LLMs into Disobedience: Understanding, Analyzing, and Preventing Jailbreaks 2023
>
> [3] Hijacking Large Language Models via Adversarial In-Context Learning. 2023
>
> [4] Data Poisoning for In-context Learning. 2024
>
>
> # Response to Experiments
>
> Thank you for your detailed feedback and for highlighting several areas where clarification is needed. We appreciate your suggestions and have addressed your concerns below.
>
> ## Response to Sample Size and Independent Trials
> Thank you for raising this important point. To ensure the robustness of our results, we have conducted three independent trials for each of the two datasets analyzed. In each trial, we randomly selected 100 examples from each task using different random seeds (42, 123, 456) to ensure unbiased selection and to capture variability in the data.
>
> Table 1 below are the results, presented as the average performance across trials with 95% confidence intervals:
>
> Table 1. Confidence Intervals of independent trials
> |  | 3 shot | 4 shot | 5 shot |
> | :--- | :--- | :--- | :--- |
> | TMW | $0.49 \pm 0.02$ | $0.50 \pm 0.02$ | $0.50+0.01$ |
> | CurDial | $0.66 \pm 0.03$ | $0.67 \pm 0.02$ | $0.72 \pm 0.03$ |
>
>
> continue...

---

> ### Author Response · Authors · 2024-11-23
> **Rebuttal by Authors [2/7]**
>
> The narrow confidence intervals indicate stable performance across different random samples, suggesting that our sample size is sufficient for the analyses conducted. We have included these details in the appendix of the revised manuscript for completeness. We hope this additional information addresses your concerns.
>
> ## Response to Experiments on Diverse LLMs
> Thank you for this valuable suggestion. In response, we have expanded our analysis to include four additional models: Mistral-7B, Gemma-7B, Llama2-7B, and Llama2-13B. While our findings could indeed extend to proprietary models, considerations of API changes over time and reproducibility have led us to focus on open-source models for this study.
>
> The results are presented in Tables 2–5, which have been added to Appendix F of the manuscript. Key findings are as follows:
>
> -   *Sensitivity to Permutations Across Different LLM Families:* We observed that different LLM families exhibit varying sensitivities to permutations. Overall, the sensitivity ranking is Llama > Gemma > Mistral. Despite this variance, the phenomenon remains significant, with performance drops exceeding 10 percentage points in most cases.
>
> -   *Effectiveness of Our Method:* For cases involving three or more shots, our method consistently achieved improvements exceeding 10% in worst-case performance, validating the effectiveness of our approach across different models.
>
> We appreciate your suggestion and believe that this expanded analysis provides a more comprehensive understanding of how the vulnerability impacts models outside of the Llama family.
>
>
> Table 2. Experiments on Llama2-7B. Average and per-task performance are reported, with performance gains (%) shown in parentheses.
> | #Shot | Method | Average | Average | CSQA | CSQA | CurDial | CurDial | CoLA | CoLA | TMW | TMW |
> | :---: | :---: | :---: | :---: | :---: | :---: | :---: | :---: | :---: | :---: | :---: | :---: |
> |  |  | Avg. | Worst. | Avg. | Worst. | Avg. | Worst. | Avg. | Worst. | Avg. | Worst. |
> | 2 | ERM | 56.6 | 46.3 | 56.0 | 50.0 | 61.3 | 50.2 | 58.2 | 42.0 | 50.7 | 43.1 |
> |  | PEARL | 57.4 (+1.5%) | 46.5 (+0.4%) | 58.0 | 48.0 | 55.2 | 44.7 | 62.0 | 48.0 | 54.4 | 45.4 |
> | 3 | ERM | 58.2 | 34.0 | 52.7 | 34.0 | 64.0 | 36.4 | 66.0 | 36.0 | 50.1 | 29.4 |
> |  | PEARL | 59.6 (+2.3%) | 40.4 (+19.1%) | 56.3 | 40.0 | 66.2 | 46.2 | 67.0 | 42.0 | 48.7 | 33.5 |
> | 4 | ERM | 58.9 | 19.9 | 60.0 | 26.0 | 68.1 | 24.4 | 60.2 | 14.0 | 47.3 | 15.1 |
> |  | PEARL | 60.5 (+2.7%) | 31.6 (+59.1%) | 61.2 | 40.0 | 69.4 | 40.1 | 62.4 | 24.0 | 48.9 | 22.4 |
> | 5 | ERM | 61.9 | 25.8 | 59.0 | 32.0 | 74.2 | 43.9 | 65.7 | 10.0 | 48.6 | 17.1 |
> |  | PEARL | 62.9 (+1.6%) | 32.1 (+24.7%) | 62.4 | 38.0 | 73.3 | 43.4 | 64.8 | 24.0 | 51.0 | 23.0 |
>
> Table 3. Experiments on Llama2-13B
> | #Shot | Method | Average | Average | CSQA | CSQA | CurDial | CurDial | CoLA | CoLA | TMW | TMW |
> | :--- | :--- | :--- | :--- | :--- | :--- | :--- | :--- | :--- | :--- | :--- | :--- |
> |  |  | Avg. | Worst. | Avg. | Worst. | Avg. | Worst. | Avg. | Worst. | Avg. | Worst. |
> | 2 | ERM | 66.3 | 56.6 | 56.0 | 46.0 | 72.6 | 56.2 | 83.0 | 76.0 | 53.4 | 48.0 |
> |  | PEARL | 67.9 (+2.4%) | 60.7 (+7.3%) | 64.0 | 58.0 | 73.8 | 64.2 | 81.0 | 76.0 | 52.6 | 44.4 |
> | 3 | ERM | 65.7 | 46.2 | 55.7 | 38.0 | 76.4 | 51.3 | 77.7 | 56.0 | 53.1 | 39.6 |
> |  | PEARL | 68.5 (+4.2%) | 50.3 (+8.7%) | 62.7 | 44.0 | 81.0 | 58.4 | 76.7 | 56.0 | 53.5 | 42.6 |
> | 4 | ERM | 65.8 | 33.2 | 58.2 | 28.0 | 79.6 | 41.6 | 73.7 | 38.0 | 51.8 | 25.0 |
> |  | PEARL | 66.4 (+0.9%) | 40.2 (+21.1%) | 63.3 | 42.0 | 80.4 | 45.5 | 69.4 | 42.0 | 53.1 | 29.1 |
>
> Table 4. Experiments on Mistral 7B v0.2
> | #Shot | Method | Average | Average | CSQA | CSQA | CurDial | CurDial | CoLA | CoLA | TMW | TMW |
> | :---: | :---: | :---: | :---: | :---: | :---: | :---: | :---: | :---: | :---: | :---: | :---: |
> |  |  | Avg. | Worst. | Avg. | Worst. | Avg. | Worst. | Avg. | Worst. | Avg. | Worst. |
> | 2 | ERM | 64.1 | 58.1 | 67.0 | 64.0 | 54.6 | 41.8 | 81.0 | 78.0 | 53.7 | 48.5 |
> |  | PEARL | 67.0 (+4.5%) | 62.4 (+7.5%) | 68.0 | 66.0 | 59.4 | 49.0 | 82.0 | 78.0 | 58.4 | 56.7 |
> | 3 | ERM | 66.6 | 56.1 | 67.0 | 62.0 | 63.7 | 38.9 | 80.0 | 76.0 | 55.6 | 47.3 |
> |  | PEARL | 69.5 (+4.3%) | 62.8 (+12.0%) | 70.0 | 66.0 | 70.1 | 60.1 | 83.6 | 78.0 | 54.1 | 47.0 |
> | 4 | ERM | 66.7 | 50.4 | 68.9 | 60.0 | 67.6 | 47.8 | 74.2 | 52.0 | 55.9 | 41.6 |
> |  | PEARL | 68.3 (+2.5%) | 57.1 (+13.4%) | 69.9 | 62.0 | 71.6 | 54.8 | 74.9 | 66.0 | 56.8 | 45.5 |
> | 5 | ERM | 67.9 | 50.7 | 67.5 | 56.0 | 70.7 | 52.6 | 76.0 | 56.0 | 57.4 | 38.2 |
> |  | PEARL | 70.2 (+3.4%) | $58.1(+14.5%)$ | 70.4 | 64.0 | 76.7 | 59.3 | 73.3 | 66.0 | 60.4 | 43.0 |
>
> continue...

---

> ### Author Response · Authors · 2024-11-23
> **Rebuttal by Authors [3/7]**
>
> Table 5. Experiments on Gemma-7B
> | #Shot | Method | Average | Average | CSQA | CSQA | CurDial | CurDial | CoLA | CoLA | TMW | TMW |
> | :---: | :---: | :---: | :---: | :---: | :---: | :---: | :---: | :---: | :---: | :---: | :---: |
> |  |  | Avg. | Worst. | Avg. | Worst. | Avg. | Worst. | Avg. | Worst. | Avg. | Worst. |
> | 2 | ERM | 66.2 | 59.5 | 71.0 | 70.0 | 59.1 | 46.1 | 77.0 | 70.0 | 57.8 | 52.0 |
> |  | PEARL | 66.3 (+0.0) | 60.7 (+2.0) | 74.0 | 68.0 | 47.3 | 39.2 | 82.0 | 78.0 | 61.7 | 57.6 |
> | 3 | ERM | 64.7 | 52.5 | 70.7 | 64.0 | 67.1 | 45.2 | 70.3 | 60.0 | 50.5 | 40.7 |
> |  | PEARL | 68.4 (+5.8) | 59.3 (+13.0) | 74.7 | 68.0 | 59.2 | 42.5 | 78.7 | 76.0 | 61.0 | 50.6 |
> | 4 | ERM | 65.0 | 46.5 | 65.0 | 54.0 | 71.4 | 41.1 | 72.5 | 58.0 | 51.1 | 32.9 |
> |  | PEARL | 67.2 (+3.4) | 52.5 (+13.0) | 71.4 | 60.0 | 60.7 | 38.9 | 75.9 | 66.0 | 60.8 | 45.2 |
> | 5 | ERM | 64.3 | 46.3 | 65.9 | 54.0 | 73.4 | 48.3 | 65.6 | 50.0 | 52.3 | 32.9 |
> |  | PEARL | 66.3 (+3.1) | 51.0 (+10.2) | 70.3 | 60.0 | 63.4 | 43.6 | 71.3 | 60.0 | 60.2 | 40.4 |
>
>
> ## Response to Questions on Threshold τ and Number of Examples n
>
> Thank you for pointing this out. You are correct that the performance drop is influenced significantly by the number of in-context examples, n. As n increases, the number of possible permutations increases factorially (n!), which raises the likelihood of encountering permutations that degrade the model's performance.
>
> The threshold τ is used in our study to determine whether an attack on a sample is considered successful. Specifically, we define an attack as successful if the performance drop exceeds τ. Therefore, τ itself does not influence the actual performance drop but serves as a criterion for measuring the impact of different permutations on the model's output.
>
> We have expanded on this explanation in the revised manuscript to clarify the roles of τ and n in our analysis.
>
> ### Response to Calculations of $\mu_i$ and $\omega_i$ in in Equation (1)
>
> Thank you for your inquiry regarding the calculations in Equation (1). We apologize for any confusion caused by the original presentation. We have revised the formulas in the manuscript for better clarity.
>
> Given a task $D={(p_i, x_i, y_i)}$,  where $p_i$ represents an ICL prompt containing n demonstrations, we define a sample $\left(p_i, x_i, y_i\right)$ as successfully attacked if its relative performance degradation induced by a attacher exceeds a threshold $\delta \in[0 \%, 100 \%]$.
>
> We denote the set of all possible permutations of the $p_i$ demonstrations as $\mathbb{P}=\\\{Pi_0, \ldots, \Pi_{n!-1}\\\}$, where $|\mathbb{P}|=n!$. Let $g$ be a performance metric function (e.g., ROUGE-L). The ASR for a dataset $D$ is defined as:
>
> $$
> \operatorname{ASR}(D, \delta)=\frac{1}{|D|} \sum_{i=1}^{|D|} \mathbb{I}\left(\frac{\mu_i-\omega_i}{\mu_i} \geq \delta\right)
> $$
>
> where $\mathbb{I}$ denotes the indicator function, $|D|$ is the size of the dataset, and $\delta$ is the threshold.
>
> - **Average Performance $\mu_i$**: The expected performance over all permutations:
>
> $$
> \mu_i=\mathbb{E}_{\Pi \sim \mathbb{P}}[g(\Pi \cdot p_i, x_i ; y_i)]
> $$
>
>
> - **Compromised Performance $\omega_i$**:  The performance under attack.
>
> *For Exhaustive Search Attacks*, the attacked performance is calculated by testing all possible permutations of demonstrations in $Q_i$ and identifying one that yields the poorest performance:
>
> $$
> \omega_i=\min _{\Pi \in \mathbb{P}} g\left(\Pi \cdot p_i, x_i ; y_i\right)
> $$
>
> *For Neural Search Attacks*, the P-Net generates the most challenging permutation $\Pi_i$ for each sample $(p_i, x_i, y_i)$, and the attacked performance is calculated as:
>
> $$
> \omega_i=g\left(\Pi_i \cdot p_i, x_i ; y_i\right), \quad s.t. \Pi_i \sim \mathrm{PNet}\left(p_i, x_i, y_i\right)
> $$
>
> We have detailed these calculations in Section 3, Equations (1-4) of our manuscript. Thank you again for pointing the ambiguity.
>
> We hope this clarification addresses your concerns regarding how $\mu_i$ and $\omega_i$ are computed.
>
> ## Response to Question on Figure 2
> We apologize for the confusion caused by Fig. 2 and its accompanying description.
>
> Fig. 2 illustrates the different behaviors of ERM and DRO on less frequent but valid permutations. To this end, we distinguish between two types of distributions:
>
> **Empirical Distribution ($\hat{P}$):** This represents the observed distribution of the data in the training corpus. For a 3-shot training example $(p,x,y)$, where prompt $p$ contains three demonstrations, there are six possible permutations, denoted as $\{(p^0, x, y), \ldots, (p^5, x, y)\}$. These permutations are indexed from 0 to 5. The bars in Fig. 2 shows the probability of each permutation occurring in the training corpus. Specifically, permutations 0, 1, and 4 appear in the training data with frequencies $\hat{P}(p^0, x, y) > \hat{P}(p^4, x, y) > \hat{P}(p^1, x, y)$, while permutations 2, 3, and 5 do not appear, hence $\hat{P}(p^2, x, y) = \hat{P}p^3, x, y) = \hat{P}(p^5, x, y) = 0$.
>
> continue...

---

> ### Author Response · Authors · 2024-11-23
> **Rebuttal by Authors [4/7]**
>
> **Learned Distribution ($P_\theta$):** This represents the distribution learned by the LM parameterized by $\theta$ over the training data. To differentiate this from the empirical distribution (represented by bars), it is represented by a curves.
> - In the **ERM** setting (left plot), the model tends to assign higher probabilities to the more frequently occurring permutations (0, 1, 4) and lower probabilities to the less frequent ones (2, 3, 5). This bias can lead to performance degradation when permutations unseen during training occur during inference.
> - In the **DRO** setting (right plot), the model is trained to perform well across all possible permutations by considering the worst-case scenario over the ambiguity set $\mathcal{Q}$ (as shown in Equation (6) in the manuscript), thereby assigning a more balanced probability across all forms of the data (0-5) and mitigating the issue of unseen permutations during inference.
>
> The left and right plots share the empirical distribution $\hat{P}$ (depicted as bars) and differ in the learned distribution $P_\theta$ (depicted as curves) influenced by different learning algorithm used ((a) ERM, (b) DRO). All instances of $P$ in the figures should be capitalized to maintain consistency.
>
> We have revised Figure 2 and its caption, as well as the corresponding text in the manuscript, to incorporate these explanations and improve clarity.
>
>
> ## Response to Hyperparameter Selection
> Thank you for highlighting the need for a clearer explanation of our hyperparameter selection. We have added a detailed description of our hyperparameter tuning process in Appendix C.3 and D of the manuscript.
>
> The analysis of the hyperparameters for P-Net, specifically those associated with Optimal Transport (OT)—temperature, iteration coefficient, and entropy-constrained coefficient—was conducted through a grid search. These hyperparameters were selected based on average validation performance. The results of this analysis were previously reported and can be found in Appendix D, "Analysis of Hyperparameters." P-Net’s other basic hyperparameters such as batch size follows common configurations and are listed in Appendix C.3.
>
>
> For the basic hyperparameters of the LLM, such as learning rate and batch size, we adopted common configurations from SFT. Furthermore, the LoRA hyperparameters were set according to the recommended settings from LlamaFactory [5]. These parameters were consistently maintained across all experimental groups to ensure a fair comparison. These are also are listed in Appendix C.3.
>
> Additionally, we plan to open-source our code and models to enable further research by the community.
>
> [5] https://github.com/hiyouga/LLaMA-Factory
>
>
> ## Response to Question on ROUGE-L
> Thank you for raising this question about the evaluation metrics. We would like to explain the calculation of ROUGE-L and justify its usage in our work.
>
> The ROUGE-L metric is calculated as follows:
>
> $$  R_L = \frac{\text{LCS}(X,Y)}{|X|} $$  $$  P_L = \frac{\text{LCS}(X,Y)}{|Y|} $$  $$  F_L = \frac{(1 + \beta^2)R_L P_L}{R_L + \beta^2 P_L} $$
>
> where:
> - $X$ is the reference text.
> - $Y$ is the hypothesis text.
> - $\text{LCS}(X,Y)$ is the length of the longest common subsequence between $X$ and $Y$
> - $|X|$ and $|Y|$ are the lengths of $X$ and $Y$, respectively.
> -  $\beta$ is a parameter that determines the relative importance of precision and recall.
>
> We chose ROUGE-L as our evaluation metric for two main reasons:
>
> 1.  *Standard Usage in Natural Instruction Benchmarks*: It is the standard evaluation metric used in Natural Instruction benchmarks (v1 and v2) [6,7], from which we sourced our instruction tuning data. This metric provides a unified evaluation framework by treating all tasks as generation tasks.
>
>
> 2.  *Strong Correlation with Classification Accuracy*: For classification tasks, ROUGE-L shows strong correlation with accuracy metrics. The Natural Instruction benchmark [4] reported a Pearson correlation coefficient of 0.97. Our independent validation on the COLA dataset (using 50 samples) showed a correlation coefficient of 0.98.
>
>
>
> We hope these explanations can address your question.
>
> [6] Cross-Task Generalization via Natural Language Crowdsourcing Instructions. ACL 2022
>
> [7] SUPER-NATURALINSTRUCTIONS: Generalization via Declarative Instructions on 1600+ NLP Tasks. EMNLP 2022

---

> ### Author Response · Authors · 2024-11-23
> **Rebuttal by Authors [5/7]**
>
> # Response to Proposed Method
>
> We appreciate your thorough review of the methodology. All content suggestions have been revised in Section 4 and are highlighted in blue for clarity.
>
> ## Response to Question on NP-Hard and the Definition of Ambiguity Set $\mathcal{Q}$
> Thank you for your questions about the NP-hardness and the definition of the ambiguity set.
>
> *Regarding NP-hardness*: We apologize for this imprecision. While search space of the combinatorial optimization problem is indeed exponential O(n!), we cannot rigorously prove that it is NP-hard. We have corrected this claim in the manuscript.
>
> *Regarding the definition of $\mathcal{Q}$:* The ambiguity set $\mathcal{Q}$ is constructed to capture all distributions obtained by permuting the prompts in the empirical distribution $\hat{P}$. Specifically, for each possible permutation $\Pi \in \mathbb{P}$, we define the permuted distribution $Q_{\Pi}$ by applying $\Pi$ to the prompt $p$ of each data point in $\hat{P}$ :
>
> $$
> Q_{\Pi}:=\\\{(\Pi \cdot p, x, y) \mid(p, x, y) \sim \hat{P}\\\}, \quad \Pi \in \mathbb{P}
> $$
>
> where $\Pi$ is a permutation matrix acting on the sequence of demonstrations in $p$, and $\mathbb{P}$ denotes the set of all possible permutation matrices. The ambiguity set $\mathcal{Q}$ is then defined as the convex hull of these permuted distributions:
>
> $$
> \mathcal{Q}:=\\\{\sum_{\Pi \in \mathbb{P}} q_{\Pi} Q_{\Pi} \mid q \in \Delta_{|\mathbb{P}|-1}\\\}
> $$
>
> where $q$ is a probability vector belonging to the $|\mathbb{P}|-1$-dimensional simplex $\Delta_{|\mathbb{P}|-1}$.
>
>
> We have updated these detailed definitions in the manu and hope they can address your concerns.
>
>
> ## Response to the [CLS] Token's Role and Function
>
> Thank you for your question. The [CLS] token is a type of artificially defined special separator first introduced in BERT [8] (which can be replaced by other special separators like [bos], [eos]), and it serves two main purposes:
>
> 1.  Acts as a separator between different demonstrations in the text sequence
> 2.  Helps learn representations for each demonstration [9]
>
>
> We hope this clarifies your question.
>
> [8] BERT: Pre-training of Deep Bidirectional Transformers for Language Understanding. NAACL 2019
>
> [9] Less is More: Pretrain a Strong Siamese Encoder for Dense Text Retrieval Using a Weak Decoder. EMNLP 2021
>
> ## Clarification about Adversarial Training
> Thank you for your thoughtful question. We would like to clarify that our problem formulation aligns with adversarial training due to its inherent min-max structure, as demonstrated in Equation (6).
>
> Additionally, the optimization objectives of the P-Net and the LLM are opposites (excluding the entropy constraint loss of the P-Net). Specifically, the P-Net maximizes the LLM's loss (Equation (17)) when the LLM parameters are fixed, whereas the LLM minimizes the LLM's loss (Equation (14)) when the P-Net parameters are fixed. This dynamic establishes a zero-sum game structure between the two components.
>
> We have updated the manuscript with a more detailed explanation in Section 4.3. We hope this clarification addresses your concern, and we greatly appreciate your feedback.
>
> ## Clarification about the OT Equations
> Thank you for your questions and for highlighting areas that require further clarification. We address each point separately:
>
> Dimensionality of X: In Equation (10), $X ∈ R^{n×n}$, where n is the number of demonstrations.  (Note: X is now replaced by R.). We have expanded this explanation in the manuscript.
>
> To clarify, the **P-Net $: (\mathcal{P} \times \mathcal{X} \times \mathcal{Y}) \rightarrow \Delta(\Pi)$, which maps an input example to a distribution of challenging permutations**. To this end, we employ **sinkhorn algorithm, whose output converges to a the distribution of permutations (a doubly stochastic matrix)**, However, we need to sample a specific permutation from this distribution while ensuring the overall process differentiable for gradient-based optimization. To achieve this, we apply **Gumbel sampling [10]. The "introduction of randomness" and "control of discreteness" refer to this sampling process**.
>
> We hope that these explanations and clarifications can address your concerns and improve the clarity of our work.
>
> [10] Categorical Reparameterization with Gumbel-Softmax. ICLR 2017
>
> ## Clarification about the Question "OT deals with distributions over perturbations or perturbations"
>
> Thank you for your question. To clarify, the OT in our work aims to transform the input permutation distribution into a target distribution that is more challenging for current LLMs. To achieve this, we design P-Net, a mapping function defined as $P\text{-Net}: (\mathcal{P} \times \mathcal{X} \times \mathcal{Y}) \rightarrow \Delta(\Pi)$, which maps an input example to a distribution over challenging permutations. Thus, we are working with distributions over perturbations, not the perturbations themselves.
>
> We hope this clarification addresses your concerns.

---

> ### Author Response · Authors · 2024-11-23
> **Rebuttal by Authors [6/7]**
>
> ## Clarification about the Question "where the sinkhorn operator factors in Equation (8)"
>
> Thank you for your questions regarding the sinkhorn operator. We address these points below:
>
> - **Clarification on the Absence of the Sinkhorn Operator in Optimization Objective (14)**: The Sinkhorn operator is indeed utilized within our methodology; however, it is embedded in the sampling process denoted by $\Pi \sim P\text{-Net}(\phi; p,x,y)$ for the sake of brevity and clarity in presentation. The Sinkhorn operator, being parameter-free, does not appear explicitly in the optimization objective where the focus is on the trainable parameters of P-Net. We have added this clarification in the manuscript.
>
> - **Connection Between Formulations (8) and (3)**: The sampling step $\Pi \sim P\text{-Net}(\phi; p,x,y)$ in our P-Net training objective formulation (15), corresponds to addressing the inner maximization problem presented in formulation (3).
>
> We have rewrite the formulas and descriptions in Section 3.4 to make it clearer, and hope they can address your concerns.
>
> ## Clarification about "trivial solutions: outputting even matrices that average out the semantic content"
>
> Thank you for your query. We clarify that P-Net outputs an approximate permutation matrix (Figure 3 and Sec. 4.2). Ideally, this should preserve the distinctiveness of each input element. However, P-Net may resort to shortcuts to producing trivial solutions, like the one described below, to artificially increase LLM's loss during adversarial training:
>
> $\left[\begin{array}{ll}[0.5, 0.5], [0.5, 0.5]\end{array}\right] \cdot\left[x_1, x_2\right]=\left[0.5 \times x_1+0.5 \times x_2, 0.5 \times x_1+0.5 \times x_2\right]$
>
> This results in semantic overlap by averaging the input features (demonstrations in our case), which leads to a loss of distinct information. While such extreme scenarios are unlikely, we are committed to ensuring that P-Net avoids these trivial solutions.
>
> We hope this explanation can address your question. We appreciate your input and are open to further discussion.
>
> ## Clarification about "the steps between Equation (9) and (10)"
> Thank you for your question. We explain the steps between (9) and (10) in details. Note that (9) and (10) now correspond to (14) and (17) respectively in the updated manuscript.
>
> We first optimize the P-Net, corresponding to the inner maximization step in Equation (6). For a given example $(p, x, y)$, we sample a permutation $\Pi \sim \operatorname{P-Net}(\phi ;(p, x, y))$ from P-Net. We then compute the LLM's loss on the permuted example $(\Pi \cdot p, x, y)$, denoted by $\ell(\theta ; \phi ;(\Pi \cdot p, x, y))$. The objective is to optimize the P-Net parameters $\phi$ to maximize this loss:
>
> $$
> L(\phi; \theta)_lm = \mathbb{E}[\ell(\theta; \phi; (\Pi \cdot p, x, y))] \quad \text{s.t. } (p, x, y) \sim \hat{P}, \Pi \sim \text{P-Net}(\phi; (p, x, y))  \quad   \quad (14)
> $$
>
> Note that the Sinkhorn operator is implicitly included in $\Pi \sim \mathrm{P}-\mathrm{Net}(\phi ;(p, x, y))$.
> To prevent the P-Net from exploiting trivial solutions, such as outputting uniform matrices that dilute the semantic content of the demonstrations, we introduce an element-wise entropy constraint term that encourages $\Pi$ to be as distinct as possible:
>
> $$
> L(\phi)_ent=\mathbb{E}[\Pi(1-\Pi)] \quad \text{s.t. } (p, x, y) \sim \hat{P}, \Pi \sim \mathrm{P}-\mathrm{Net}(\phi ;(p, x, y))  \quad   \quad (15)
> $$
>
>
> This leads to the following combined optimization for the P-Net:
>
> $$
> \hat{\phi}^{\star}=\arg \max _{\phi \in \Phi}(L(\phi ; \theta)_lm-\beta L(\phi)_ent)  \quad   \quad (16)
> $$
>
>
> where $\beta$ represents the penalty coefficient for the entropy constraint.
>
> ***Note when optimizing Equation (16), $\theta$ remains constant.***
>
> We then optimize LLM, corresponding to the inner minimization step in Equation (6). For an example ( $p, x, y)$, we get a challenging permutation from the previously optmized $P$-Net $\left(\hat{\phi}^{\star}\right), \Pi \sim$ $P$-Net $\left(\hat{\phi}^{\star} ;(p, x, y)\right)$. We compute the LLM's loss on this permuted example $(\Pi \cdot p, x, y)$, denoted by $\ell\left(\theta ; \hat{\phi}^{\star} ;(\Pi \cdot p, x, y)\right)$. The objective is to optimize the LLM parameters $\theta$ to minimize this loss:
>
> $$
> \hat{\theta}^{\star}=\arg \max _{\theta \in \Theta} L\left(\hat{\phi}^{\star} ; \theta\right)_lm,  \quad   \quad (17)
> $$
>
>
> ***Note when optimizing LLM, we incorporate the previously optimized parameter $\hat{\phi}^{\star}$ from the P-Net and keep it constant.***
> From Equation (16) to (17), we complete a loop of iteration. In the next iteration, we substitute $\hat{\theta}^{\star}$ into Equation (16) for a new round of optimization until convergence.
>
> We hope these explanation can address your questions and look forward to any additional comments you may have.

---

> ### Author Response · Authors · 2024-11-23
> **Rebuttal by Authors [7/7]**
>
> ## Response to Related Work
> Thank you for these thoughtful suggestions regarding related work. We have strengthened the manuscript with following additions:
>
> - We added relevant learning theory citations in Section 4 to provide more context for interested readers.
>
> - We have incorporated discussion of OOD Generlization and GroupDRO work into the DRO paragraph in related work section.
>
> These additions will provide readers with important context and enable them to explore the theoretical underpinnings in more depth. We appreciate the guidance on improving the manuscript's scholarly rigor.
>
> ## Response to Miscellaneous
>
> Thank you for these points.
>
> -   "General performance" refers to average performance or performance with a random permutation, both of which are now reported in Figure 1 as suggest.
>
> -   Modified Figure 1 with a more distinctive color palette to improve readability.
>
> ## Response to Question 1
>
> Thank you for your question. Exploring the best-case performance scenario is indeed intriguing. Although our methodology was initially designed to optimize for pessimistic (worst-case) scenarios, we have also included an evaluation of the best-case performance for both PEARL and ERM to provide a balanced perspective. The results are shown in the Table 6 below.
>
> Table 6: Best performance comparison between ERM and PEARL.
> | #Shot | Method | Average | Average | CSQA | CurDial | CoLA | TMW |
> | :---: | :---: | :---: | :---: | :---: | :---: | :---: | :---: |
> |  |  | Best | gain | Best | Best | Best | Best |
> | 2.0 | ERM | 64.1 |  | 68.8 | 64.4 | 64.1 | 59.2 |
> |  | PEARL | 68.8 | 7.2% | 73.4 | 69.2 | 70.3 | 62.1 |
> | 3.0 | ERM | 72.8 |  | 70.3 | 85.0 | 65.6 | 70.3 |
> |  | PEARL | 77.0 | 5.7% | 73.4 | 87.9 | 79.7 | 66.9 |
> | 4.0 | ERM | 82.9 |  | 81.3 | 92.4 | 78.1 | 79.7 |
> |  | PEARL | 84.3 | 1.7% | 82.8 | 93.6 | 81.2 | 79.5 |
> | 5.0 | ERM | 86.8 |  | 84.4 | 95.3 | 81.3 | 86.2 |
> |  | PEARL | 89.3 | 2.9% | 87.5 | 96.5 | 85.9 | 87.3 |
>
> Surprisingly, the results, show that across all datasets and in every shot condition, PEARL's best performance consistently exceeded that of ERM, although the overall average improvement was modest.
>
> We appreciate your insightful suggestion, as it adds a significant dimension to our model evaluation. These findings have been included in the appendix for reference.
>
> Thank you once again for your thoughtful engagement.
>
> ## Response to Question 2
> Thank you for this suggestion. We have:
>
> -   Added the random permutation baseline to Figure 1 and change the color.
>
> -   The average performance calculation method was previously explained in our response and in Equation (3).
>
> ## Response to Question 3
>
> Thank you for the query. This is indeed the actual upper bound, as previously clarified in our response and in added Equation (3).
>
>
>
> ----------
>
> We sincerely appreciate your detailed and insightful comments, which have undoubtedly strengthened our work. We hope that our responses and the updated manuscript effectively address your concerns and improve the clarity of our contribution. We look forward to any additional feedback you may have.

---

> ### Author Response · Authors · 2024-11-25
> **Gentle Reminder: Request for Your Feedback on Our Response**
>
> Dear Reviewer n5jf,
>
> We hope this message finds you well.
>
> This is a kind reminder that the discussion phase of the review process is ending soon.
>
> We have carefully addressed your insightful comments and suggestions in our response, including detailed explanations, discussions, and new experimental results. Based on your feedback, we have made detailed revisions to the manuscript, particularly in Sections 3 and 4, to clarify the threat model and enhance the description of our proposed method.
>
> Your feedback is crucial in helping us confirm whether we have appropriately addressed the issues and suggestions you raised. We would greatly appreciate any updates or further feedback from you. Your insights and expertise are invaluable to us, as we aim to address any outstanding concerns and improve our submission.
>
> Thank you for your time and attention. We look forward to hearing from you soon.
>
> Best regards,
>
> Authors

---

> ### Author Response · Authors · 2024-11-27
> **Gentle Reminder: Request for Your Feedback**
>
> Dear Reviewer n5jf,
>
> As today is the deadline for our manuscript revisions, we kindly seek your feedback on our responses and the revised manuscript. We deeply appreciate the time and effort you invested in providing detailed and thoughtful comments on our work.
>
> In return, we have worked diligently to address each of your concerns point-by-point, culminating in a comprehensive 7-page response along with careful revisions to the manuscript. We sincerely hope these efforts align with your expectations and address your concerns thoroughly.
>
> We would be truly grateful if you could review our responses in your busy time and let us know if they sufficiently address your feedback. Your input is invaluable to us.
>
> Best regards,
>
> Authors

---

> ### Author Response · Authors · 2024-11-29
> **Request for feedback on our rebuttal**
>
> Dear Reviewer n5jf,
>
> We noticed that we haven't yet received your feedback on our submitted response to your review. As the discussion phase is ending in the next three days, we would particularly appreciate your thoughts on our clarifications.
>
>  In our response, we have:
>
> 1. **Thoroughly clarified your main concern about the practicality of our threat model setting**: We have explained that our model follows the standard two-party setting (between an attacker and a defender) widely considered in the field of adversarial attacks for LLMs, which does not require network interception techniques as in a three-party setting. (Modified Section 3)
>
> 2. **Provided detailed explanations for other technical questions** you raised regarding adversarial training, DRO, Gumbel sampling and OT techniques. (Modified Sections 3 and 4)
>
> 3. **Added new experimental results and made other necessary revisions** as suggested. (Modified Section 3, Added Appendices E and G)
>
> For your convenience, we have highlighted these changes in blue in the manuscript.
>
> We believe we have addressed your primary concerns and would greatly value your feedback on these points if there are any outstanding questions.
>
> Best regards,
>
> Authors

---

> ### Author Response · Authors · 2024-12-02
> **Gentle Reminder: Request for Your Feedback**
>
> Dear Reviewer n5jf,
>
> This is a kind reminder that the discussion phase of the review process is ending soon.
>
> Thank you for your time and initial review. We look forward to hearing from you.
>
> Best regards,
>
> Authors

---

> ### Author Response · Authors · 2024-12-04
> **Gentle Note: Looking Forward to Your Feedback**
>
> Dear Reviewer n5jf,
>
> We notice that we haven't received your feedback on our response yet. We look forward to hearing from you.
>
> Best regards,
>
> Authors

---

### Official Review · Reviewer_cDtR · 2024-10-29

**Soundness:** 3
**Presentation:** 3
**Contribution:** 2
**Rating:** 8
**Confidence:** 3

**Summary:**

The paper address the problem that the order of demonstrations in few-shot learning influences the task performance. The authors propose an adversarial optimisation training scheme, whereby an additional model mines hard permutations while fine-tuning the original model to perform well on them.

**Strengths:**

1. The paper offers a simple, straightforward, intuitive and, most importantly, well-performing solution to the problem of permutation-sensitivity.
2. The paper is clearly written and well-presented.
3. It is not immediately obvious how one can handle the combinatorial explosion for the hard permutation mining with a neural network but the authors propose an elegant solution using the Sinkhorn operator and Gumbel sampling.

**Weaknesses:**

1. The evaluations seem to be restricted to the 3-, 4- and 5-shot cases. However, these settings are quite small and one could even enumerate them and try each permutation without needing the P-Net at all. Currently, models have very large contexts and can have hundreds if not thousands of demonstrators (e.g., Many-Shot In-Context Learning, Agarwal et al., 2024). It is not clear from the paper whether such larger sets of demonstrations also exhibit such permuntation-sensitivity. Furthermore, it is not clear whether the proposed approach can scale to the many-shot setting (or more than 5 examples) and what the benefit would be in that case. I think a key question that needs to be answered is whether the proposed method is better than simply adding extra demonstrators (in random order).
2. The paper does not compare how the proposed method fares against baselines such as selecting the most performant permutation (without specially fine-tuning the model to be permutation-invariant) and selecting the demonstrators that result in maximal performance, e.g. the (Lu et al., 2022) reference from the paper.

**Questions:**

1. The separator line between 2-shot and 3-shot in Table 3 is offsetted.
2. Do you need a separate P-Net for every number of demonstrators? Eq. (4) seems to hint at that. If that's the case, I think the authors should discuss the implications of this approach on scalability and practical implementation.
3. If one wants demonstrator-wise permutation invariance, can’t one just mask the examples in a way that no example can attend to another example, and thus making the model permutation-invariant by design?

---

> ### Author Response · Authors · 2024-11-23
> **Rebuttal by Authors [1/2]**
>
> Thank you for your thoughtful and positive feedback on our work. We are delighted that you find our solution to be both intuitive and well-performing. Your acknowledgment of the elegance of our approach to handling the combinatorial explosion—using the Sinkhorn operator and Gumbel sampling—is particularly encouraging. We are also pleased that you found our paper to be clearly written and well-presented. We will address your comments and supplement our work with new experimental results.
>
> ### Response to W1: Experiments on Many-shot Setting
>
> Thank you for your suggestion. In response, we expanded our evaluations to 8, 16，32, and 64 shots. We were unable to test with 128 shots as most sequences exceeded the maximum sequence length of 8k for Llama3. The results for average and worst-case performance gains are reported in Table 1 below.
>
> Table 1: Scaling to many-shot ICL.
> | Shots | Average Gain (%) | Worst Gain (%) |
> | :---: | :--------------: | :------------: |
> | 8     | 7.6              | 39.2           |
> | 16    | 5.3              | 23.7           |
> | 32    | 3.8              | 36.4           |
> | 64    | 3.5              | 36.7           |
>
> As shown in Table 1, our method, PEARL, achieves notable worst-case performance gains ranging from 24% to 40% when generalizing to larger shot numbers (up to 64 shots) and longer sequences (up to 8k). This indicates that PEARL helps LLMs learn robust features that generalize well to many-shot ICL settings. Detailed results and visualizations have been updated in Section 6.2 and Appendix G of our manuscript, and these updates are distinctly marked in orange for easy identification.
>
> Regarding "whether the proposed method is better than simply adding extra demonstrators," this can be seen from Figure 1 and Table 2 in the manu. As the number of shots increases, the worst performance generally shows a monotonic decline because more shots lead to more permutation variants, which are likely to further reduce worst-case performance. In contrast, our method is able to improve the worst-case performance.
>
> We hope these additional results substantiate the generalizability of our method and adequately address your concerns.
>
>
> ### Response to W2: Comparison With Best Ordering Performance
> Thank you for your question. Exploring the best-case performance scenario is indeed intriguing. Although our methodology was initially designed to optimize for pessimistic (worst-case) scenarios, we have also included an evaluation of the best-case performance for both PEARL and ERM to provide a balanced perspective. The results are shown in the Table 2 below.
>
> Table 2: Best performance comparison between ERM and PEARL.
> | #Shot | Method | Average | Average | CSQA | CurDial | CoLA | TMW |
> | :---: | :---: | :---: | :---: | :---: | :---: | :---: | :---: |
> |  |  | Best | gain | Best | Best | Best | Best |
> | 2.0 | ERM | 64.1 |  | 68.8 | 64.4 | 64.1 | 59.2 |
> |  | PEARL | 68.8 | 7.2% | 73.4 | 69.2 | 70.3 | 62.1 |
> | 3.0 | ERM | 72.8 |  | 70.3 | 85.0 | 65.6 | 70.3 |
> |  | PEARL | 77.0 | 5.7% | 73.4 | 87.9 | 79.7 | 66.9 |
> | 4.0 | ERM | 82.9 |  | 81.3 | 92.4 | 78.1 | 79.7 |
> |  | PEARL | 84.3 | 1.7% | 82.8 | 93.6 | 81.2 | 79.5 |
> | 5.0 | ERM | 86.8 |  | 84.4 | 95.3 | 81.3 | 86.2 |
> |  | PEARL | 89.3 | 2.9% | 87.5 | 96.5 | 85.9 | 87.3 |
>
> As shown in Table 6, across all datasets and for every shot condition, PEARL's best-case performance consistently exceeds that of ERM. These results indicate that PEARL not only significantly improves the worst-case performance but also achieves stable improvements in the best-case performance.
>
> We have included these findings in Appendix G of our paper to provide a more comprehensive evaluation of our model.
>
> Thank you once again for your thoughtful engagement and valuable suggestion.
>
> ### Response to Q1: Formatting Corrections
> Thank you for your careful review and for pointing out the inaccuracies. We have updated the manuscript accordingly. Your meticulous feedback has undeniably enhanced the clarity of our presentation.
>
>
> ### Response to Q2: Clarification on P-Net's Adaptability to Various Shot Sizes
> Thank you for your question. To clarify, our P-Net does not require a separate model for each number of demonstrators. It is designed to handle varying numbers of shots effectively within a single model.
>
> This capability stems from its underlying transformer architecture, which is adept at processing sequences of variable lengths. In the context of P-Net, the number of shots provided corresponds to the sequence length that the transformer processes. As shown in Equations (9) and (10) of our paper, increasing the number of shots does not change the number of parameters in the model. This flexibility enhance its scalability and practical implementation.
>
> We appreciate your thoughtful question and hope this clarifies the adaptability of P-Net.

---

> ### Author Response · Authors · 2024-11-23
> **Rebuttal by Authors [2/2]**
>
> ### Response to Q3: Regarding the Architecture Improvements
> Thank you for your insightful question. Your suggestion is indeed an interesting idea. However, implementing demonstrator-wise permutation invariance by masking examples so that no example can attend to another is challenging within the current Transformer architecture. Transformers rely on autoregressive attention modules present in every layer, making it difficult to completely isolate samples from one another. Additionally, preventing interactions between demonstrations may negatively impact the model's overall performance.
>
> There is work exploring alternative architectures to address this issue. Preliminary research by (Chen et al. (2023)) shows that the DeepSet architecture exhibits better permutation invariance than Transformers. However, this MLP-based architecture is currently too limited to handle complex language modeling tasks. We have included this work in our related work section.
>
> We hope this clarifies your concern.
>
>
>
> *Reference*
>
> [1] Chen et al., 2023, Positional Information Matters for Invariant In-Context Learning: A Case Study of Simple Function Classes
>
> ***
> Thank you once again for your insightful comments, which has undoubtedly strengthened our work. We have updated our manuscript to reflect these changes, which are highlighted in orange for easy identification. We hope that our responses could effectively address your concerns, and we look forward to  any further feedback you may have.

---

> > ### Comment · Reviewer_cDtR · 2024-11-27
> >
> > I would like to thank the authors for the additional experiments with larger sequence lengths and for comparing ERM and PEARL for the best-case scenario. I also appreciate their clarification on the P-Net's adaptability to various shot sizes.
> >
> > I have a couple outstanding comments for the authors' consideration.
> >
> > How do you actually measure the worst-case performance if you can’t do an exhaustive search? Is it using the hard example mining model? But then how do you know you know that you are actually improving worst case performance vs preventing the hard example mining model from finding low performing permutations?
> >
> > > Regarding "whether the proposed method is better than simply adding extra demonstrators," this can be seen from Figure 1 and Table 2 in the manu. As the number of shots increases, the worst performance generally shows a monotonic decline because more shots lead to more permutation variants, which are likely to further reduce worst-case performance. In contrast, our method is able to improve the worst-case performance.
> >
> > I feel that the authors might have misunderstood me. While I agree that the more samples you have, the worse the worst performance will be, I would expect that the average performance would improve. I can formulate the question as “given an n-shot PEARL-trained model, how many shots (m>n) do I need to provide to a model without PEARL to achieve a similar performance”? In other words, an n-shot PEARL model is equivalent to an m-shot non-PEARL model for what m as a function of n?
> >
> > > Thank you for your insightful question. Your suggestion is indeed an interesting idea. However, implementing demonstrator-wise permutation invariance by masking examples so that no example can attend to another is challenging within the current Transformer architecture. Transformers rely on autoregressive attention modules present in every layer, making it difficult to completely isolate samples from one another. Additionally, preventing interactions between demonstrations may negatively impact the model's overall performance.
> >
> > I don’t see how the masking would be difficult here. You’d effectively have triangular matrices on the diagonal, each corresponding to one demonstrator, and with the final question part attending to everything before it. Autoregression shouldn’t matter here as presumably you are processing the demonstrators in parallel and it does not invalidate the proposed masking as it is still fully causal. I do agree that later demonstrators attending to previous ones could indeed be beneficial. Furthermore, it is not clear how the positional encodings should be handled with such masking. But nevertheless, maybe a much simpler architecture that could potentially have a similar effect as PEARL.
> >
> > Nevertheless, in light of the additional experiments and the authors' responses, I will increase my score.

---

> > > ### Author Response · Authors · 2024-11-27
> > > **Thank You for Your Recognition and Further Feedback**
> > >
> > > We sincerely thank you for recognizing our work and for continuously providing high-quality suggestions to enhance our research. We will address your points in order.
> > >
> > > ### Response to "Measuring Worst-Case Performance Without Exhaustive Search"
> > >
> > > > “How do you actually measure the worst-case performance if you can’t do an exhaustive search? Is it using the hard example mining model? But then how do you know you know that you are actually improving worst case performance vs preventing the hard example mining model from finding low performing permutations?”
> > >
> > > We apologize for any confusion caused. To clarify, in all experiments presented in Sections 5 and 6, we evaluate the worst-case performance through **exhaustive search** to ensure that we identify the true worst-case scenarios. During training, we use the hard example mining model (P-Net) to efficiently and approximately learn to generate the worst permutations. This approach allows us to focus on potential worst-case permutations during training while ensuring accurate evaluation through exhaustive search. We have added explainations in the manuscript.
> > >
> > > ### Response to "Relationship Between PEARL and Non-PEARL Models in Terms of Shot Efficiency"
> > >
> > > > “I feel that the authors might have misunderstood me... ‘Given an n-shot PEARL-trained model, how many shots (m > n) do I need to provide to a model without PEARL to achieve a similar performance?’ In other words, an n-shot PEARL model is equivalent to an m-shot non-PEARL model for what m as a function of n?”
> > >
> > >
> > > Thank you for bringing up this insightful point. We now understand that you are suggesting comparing the average performance, acknowledging that while the worst-case performance decreases as the number of shots increases, the average performance may improve. This is an excellent aspect that we had previously overlooked.
> > >
> > > As illustrated in Table 3, we compare the average performance with and without using PEARL. In most cases, an **n**-shot PEARL-trained model achieves similar average performance to a non-PEARL model with approximately **2n to 4n** shots, and sometimes even more than 4n. This demonstrates that our method is more sample-efficient, saving about 50%–75% of the sequence length (number of shots) needed to achieve the same performance. This reduction significantly enhances computational efficiency by reducing computational complexity from O(N²) to O((N/2)²) or O((N/4)²).
> > >
> > > We have presented these results and the discussion in Appendix H of our manuscript.
> > >
> > > Table 3: Average performance with and without PEARL.
> > > | # Shots | $\mathbf{2}$ | $\mathbf{4}$ | $\mathbf{8}$ | 16 | 32 | 64 |
> > > | :--- | :--- | :--- | :--- | :--- | :--- | :--- |
> > > | Without PEARL | 57.3 | 59.7 | 61.8 | 66.9 | 67.4 | 68.1 |
> > > | With PEARL | 62.9 | 63.1 | 66.5 | 70.5 | 70.0 | 70.4 |
> > >
> > >
> > > ### Implementing Permutation Invariance Transformer via Masking
> > >
> > > > “I don’t see how the masking would be difficult here. You’d effectively have triangular matrices on the diagonal, each corresponding to one demonstrator, and with the final question part attending to everything before it... I do agree that later demonstrators attending to previous ones could indeed be beneficial. Furthermore, it is not clear how the positional encodings should be handled with such masking. But nevertheless, maybe a much simpler architecture that could potentially have a similar effect as PEARL.”
> > >
> > > You are correct; such a structural modification is indeed feasible. Upon further careful consideration, we identified there are two potential drawbacks to implementing structural modifications compared to algorithmic modifications:
> > >
> > > 1.  **Robustness-Performance Trade-off:** While masking can improve permutation robustness by isolating demonstrations, it may reduce average performance. Preventing interactions between demonstrations could limit the model's ability to learn from shared patterns, which may be a major limiting factor.
> > >
> > > 2.  **Discrepancy Introduced by Modifying the Attention Mechanism:** Implementing such masking during fine-tuning requires altering the attention mechanism, introducing a non-trivial discrepancy between pretraining and fine-tuning stages. This may result in performance loss due to changes or loss of the pre-trained learning patterns.
> > >
> > > Our training algorithm-based approach avoids these issues by enhancing permutation robustness without modifying the underlying model architecture or compromising average performance. Nevertheless, we agree that exploring architectural modifications is an exciting direction for future research. We appreciate your further clarification and discussion and believe they would inspire further advancements in this area.
> > >
> > > Once again, thank you for your detailed and inspiring comments. They have undoubtedly helped us improve the quality of our work and deepen our understanding. We hope that our responses could address your concerns effectively, and look forward to receiving any further feedback you may have.

---

> > > > ### Comment · Reviewer_cDtR · 2024-11-27
> > > >
> > > > Thank you for the detailed response! Evaluating via exhaustive search makes sense. It's also really nice to see the sample efficiency view of your work as well. Though, as a petty point, O((N/4)²)=O(N²) so you are not actually improving the asymptotic complexity, but of course, we work with small number of samples where the improvements should be noticable. This is a pretty good work and I appreciate your responses to my concerns. I wish you all the best with the rest of the review process!

---

> > > > > ### Author Response · Authors · 2024-11-28
> > > > >
> > > > > We truly appreciate the insights and expert suggestions you brought to our manuscript throughout the review process. Thank you again for your recognition and kind wishes for our work.
> > > > >
> > > > > Best regards,
> > > > >
> > > > > Authors

---

### Official Review · Reviewer_ZWPA · 2024-11-02

**Soundness:** 3
**Presentation:** 4
**Contribution:** 3
**Rating:** 8
**Confidence:** 4

**Summary:**

The paper proposes a permutation-resilient learning framework (PEARL) aimed at improving the robustness of Large Language Models (LLMs) to the order in which examples are presented in in-context learning tasks. The framework is based on adversarial fine-tuning, where the adversary of the LLM tries to find a permutation of the inputs most detrimental to the LLM’s performance.

The paper starts by empirically establishing that, for Llama 3 8B, there is indeed a significant variation of in-context learning performance depending on the ordering of the inputs. Then, they propose adopting a Distributionally Robust Optimization (DRO) framework for fine-tuning LLMs to mitigate this.

In their setting, DRO requires finding permutations where the LLM performs poorly. Since an exhaustive search would be intractable, the authors propose using a neural network (referred to as P-Net) to propose hard permutations. The network outputs are then passed through the Sinkhorn algorithm, which converts them into a doubly-stochastic matrix, and a differentiable sampling step (which uses Gumbel noise), yielding a permutation matrix. The authors then propose an adversarial optimization routine for jointly training the P-net and fine-tuning the LLM. The P-net is used only for training, and can be discarded at inference time.

The authors then evaluate PEARL in the established toy setting of in-context learning of linear functions, and in a real-world LLM fine-tuning setting. In both settings, they find that PEARL improves both average- and worst-case model performance, compared to existing approaches such as raw empirical risk minimization (ERM) or ERM combined with curriculum learning.

**Strengths:**

1. Empirical results on LLM fine-tuning: the results indicating improved average- and worst-case performance in standard ICL benchmarks indicate that PEARL can be useful for practitioners as a part of their fine-tuning pipeline.

1. Effectiveness in the few-shot setting: In addition, the gains above are already noticeable with a small number of shots (2, 3 or 4), indicating the method does not require a very large number of in-context examples to be beneficial (if this were the case, it could hinder the method’s practical usefulness).

1. Practical approach to working with permutations: the proposed method for sampling hard permutations is elegant, as it reduces an intractable search problem to learning an attention-like interaction between token embeddings of ICL examples, followed by a differentiable sampling routine involving the Sinkhorn algorithm. The adversarial optimization routine enables the P-net to be discarded at inference time, not requiring any pre- or post-processing of model inputs and outputs. Meanwhile, many adversarial robustness methods do incur additional inference-time overhead.

1. Clarity and presentation: the paper is clearly written and easy to follow, despite minor points referenced below.

**Weaknesses:**

1. The authors work only with Llama 3 8B. It would be relevant to assess the model’s generalizability to include other models; in particular Llama 2 7B and 13B (from the previous generation), Mistral 7B v0.2 and Gemma 7B.

1. Lack of evaluations for many-shot settings: recent work in adversarial robustness has highlighted the vulnerability of LLMs to many-shot adversarial attacks (Anil et al. 2024). Hence, it would strengthen the paper to include additional evaluations where the number of shots is much larger than 5 (e.g. 32, 64 and 128 shots). In particular, the setup used by Anil et al. 2024 could enable the authors to derive direct consequences of their work when it comes to model safety. In addition, the many-shot setting is also relevant for agentic workflow and RAG applications. Hence, including such evaluations could further enhance the applicability of the findings for practitioners.

1. Unclear connection with Optimal Transport: the authors use the Sinkhorn algorithm, but, as someone without a background in optimal transport, it is unclear that this alone qualifies the method as learning an optimal transport map between the uniform distribution on permutations and the “distribution of hard permutations” (for which a precise definition is not given). It would be helpful if the authors could clarify this, e.g. in the appendix.

1. Minor points about formatting:

    - The use of the absolute value in Equation (1) confused me at first, as it seemed to suggest that the ASR metric also factors in cases where permuting the inputs increases performance. This is of course not the case, since the worst-case performance is never greater than the average case performance. Still, I believe it would improve clarity to remove the absolute value in Equation (1), since its argument is anyways always non-negative.

    - In Algorithm 1, the authors refer to “ascending the gradient” of functions labeled $L_D$ and $L_G$. I found this confusing, as the naming of the functions seems to suggest these are loss functions, in which case one would descend their gradients.

    - Also, the nomenclature for the functions in 4.3 is different from that of Algorithm 1. I believe it would help clarity if these matched.

**Questions:**

1. As presented in the methods section, the matrix W does not need to be positive definite, meaning that $(h_1, h_2) \mapsto h_1^T W h_2$ need not define an inner product in the model’s embedding space. Do you have an intuition or empirical results as to whether constraining W to be positive would harm performance?

1. Do you have an explanation for why the ERM+DS and ERM+IM baselines make average-case performance worse across all numbers of shots in Table 3?

---

> ### Author Response · Authors · 2024-11-23
> **Rebuttal by Authors [1/3]**
>
> Thank you for your insightful and positive feedback on our work. We are delighted that you find our proposed method, PEARL, to be both elegant and practical for practitioners as part of their fine-tuning pipeline. Your recognition of the noticeable gains in both average and worst-case performance is highly encouraging. We appreciate your acknowledgment that our approach does not incur any additional inference-time overhead. Additionally, we are pleased that you find the clarity and presentation of our paper to be strong. We will address your comments sequentially and supplement our work with new experimental results and discussions.
>
> ### Response to W1: Experiments on Diverse LLMs
>
> Thank you for your suggestion. In response, we expanded our experiments to include four additional models: Llama 2 7B and 13B (from the previous generation), Mistral 7B v0.2, and Gemma 7B. The results are presented in Tables 1-4 and have been added to Appendix F of the manuscript. Key Discoveries are as follows:
>
> -   *Sensitivity to Permutations Across Different LLM Families:* We observed that different LLM families exhibit varying sensitivities to permutations. Overall, the sensitivity ranking is Llama > Gemma > Mistral. Despite this variance, the phenomenon remains significant, with performance drops exceeding 10 percentage points in most cases.
>
> -   *Effectiveness of Our Method:* For cases involving three or more shots, our method consistently achieved improvements exceeding 10% in worst-case performance. This validates the effectiveness of our approach.
>
> We appreciate your suggestion and hope that the expanded results can addresses your concerns.
>
> Table 1. Experiments on Llama2-7B. Average and per-task performance are reported, with performance gains (%) shown in parentheses.
> | #Shot | Method | Average | Average | CSQA | CSQA | CurDial | CurDial | CoLA | CoLA | TMW | TMW |
> | :---: | :---: | :---: | :---: | :---: | :---: | :---: | :---: | :---: | :---: | :---: | :---: |
> |  |  | Avg. | Worst. | Avg. | Worst. | Avg. | Worst. | Avg. | Worst. | Avg. | Worst. |
> | 2 | ERM | 56.6 | 46.3 | 56.0 | 50.0 | 61.3 | 50.2 | 58.2 | 42.0 | 50.7 | 43.1 |
> |  | PEARL | 57.4 (+1.5) | 46.5 (+0.4) | 58.0 | 48.0 | 55.2 | 44.7 | 62.0 | 48.0 | 54.4 | 45.4 |
> | 3 | ERM | 58.2 | 34.0 | 52.7 | 34.0 | 64.0 | 36.4 | 66.0 | 36.0 | 50.1 | 29.4 |
> |  | PEARL | 59.6 (+2.3) | 40.4 (+19.1) | 56.3 | 40.0 | 66.2 | 46.2 | 67.0 | 42.0 | 48.7 | 33.5 |
> | 4 | ERM | 58.9 | 19.9 | 60.0 | 26.0 | 68.1 | 24.4 | 60.2 | 14.0 | 47.3 | 15.1 |
> |  | PEARL | 60.5 (+2.7) | 31.6 (+59.1) | 61.2 | 40.0 | 69.4 | 40.1 | 62.4 | 24.0 | 48.9 | 22.4 |
> | 5 | ERM | 61.9 | 25.8 | 59.0 | 32.0 | 74.2 | 43.9 | 65.7 | 10.0 | 48.6 | 17.1 |
> |  | PEARL | 62.9 (+1.6) | 32.1 (+24.7) | 62.4 | 38.0 | 73.3 | 43.4 | 64.8 | 24.0 | 51.0 | 23.0 |
>
> Table 2. Experiments on Llama2-13B
> | #Shot | Method | Average | Average | CSQA | CSQA | CurDial | CurDial | CoLA | CoLA | TMW | TMW |
> | :--- | :--- | :--- | :--- | :--- | :--- | :--- | :--- | :--- | :--- | :--- | :--- |
> |  |  | Avg. | Worst. | Avg. | Worst. | Avg. | Worst. | Avg. | Worst. | Avg. | Worst. |
> | 2 | ERM | 66.3 | 56.6 | 56.0 | 46.0 | 72.6 | 56.2 | 83.0 | 76.0 | 53.4 | 48.0 |
> |  | PEARL | 67.9 (+2.4) | 60.7 (+7.3) | 64.0 | 58.0 | 73.8 | 64.2 | 81.0 | 76.0 | 52.6 | 44.4 |
> | 3 | ERM | 65.7 | 46.2 | 55.7 | 38.0 | 76.4 | 51.3 | 77.7 | 56.0 | 53.1 | 39.6 |
> |  | PEARL | 68.5 (+4.2) | 50.3 (+8.7) | 62.7 | 44.0 | 81.0 | 58.4 | 76.7 | 56.0 | 53.5 | 42.6 |
> | 4 | ERM | 65.8 | 33.2 | 58.2 | 28.0 | 79.6 | 41.6 | 73.7 | 38.0 | 51.8 | 25.0 |
> |  | PEARL | 66.4 (+0.9) | 40.2 (+21.1) | 63.3 | 42.0 | 80.4 | 45.5 | 69.4 | 42.0 | 53.1 | 29.1 |
>
> Table 3. Experiments on Mistral 7B v0.2
> | #Shot | Method | Average | Average | CSQA | CSQA | CurDial | CurDial | CoLA | CoLA | TMW | TMW |
> | :---: | :---: | :---: | :---: | :---: | :---: | :---: | :---: | :---: | :---: | :---: | :---: |
> |  |  | Avg. | Worst. | Avg. | Worst. | Avg. | Worst. | Avg. | Worst. | Avg. | Worst. |
> | 2 | ERM | 64.1 | 58.1 | 67.0 | 64.0 | 54.6 | 41.8 | 81.0 | 78.0 | 53.7 | 48.5 |
> |  | PEARL | 67.0 (+4.5) | 62.4 (+7.5) | 68.0 | 66.0 | 59.4 | 49.0 | 82.0 | 78.0 | 58.4 | 56.7 |
> | 3 | ERM | 66.6 | 56.1 | 67.0 | 62.0 | 63.7 | 38.9 | 80.0 | 76.0 | 55.6 | 47.3 |
> |  | PEARL | 69.5 (+4.3) | 62.8 (+12.0) | 70.0 | 66.0 | 70.1 | 60.1 | 83.6 | 78.0 | 54.1 | 47.0 |
> | 4 | ERM | 66.7 | 50.4 | 68.9 | 60.0 | 67.6 | 47.8 | 74.2 | 52.0 | 55.9 | 41.6 |
> |  | PEARL | 68.3 (+2.5) | 57.1 (+13.4) | 69.9 | 62.0 | 71.6 | 54.8 | 74.9 | 66.0 | 56.8 | 45.5 |
> | 5 | ERM | 67.9 | 50.7 | 67.5 | 56.0 | 70.7 | 52.6 | 76.0 | 56.0 | 57.4 | 38.2 |
> |  | PEARL | 70.2 (+3.4) | $58.1(+14.5)$ | 70.4 | 64.0 | 76.7 | 59.3 | 73.3 | 66.0 | 60.4 | 43.0 |
>
>
> continued...

---

> ### Author Response · Authors · 2024-11-23
> **Rebuttal by Authors [2/3]**
>
> Table 4. Experiments on Gemma-7B
> | #Shot | Method | Average | Average | CSQA | CSQA | CurDial | CurDial | CoLA | CoLA | TMW | TMW |
> | :---: | :---: | :---: | :---: | :---: | :---: | :---: | :---: | :---: | :---: | :---: | :---: |
> |  |  | Avg. | Worst. | Avg. | Worst. | Avg. | Worst. | Avg. | Worst. | Avg. | Worst. |
> | 2 | ERM | 66.2 | 59.5 | 71.0 | 70.0 | 59.1 | 46.1 | 77.0 | 70.0 | 57.8 | 52.0 |
> |  | PEARL | 66.3 (+0.0) | 60.7 (+2.0) | 74.0 | 68.0 | 47.3 | 39.2 | 82.0 | 78.0 | 61.7 | 57.6 |
> | 3 | ERM | 64.7 | 52.5 | 70.7 | 64.0 | 67.1 | 45.2 | 70.3 | 60.0 | 50.5 | 40.7 |
> |  | PEARL | 68.4 (+5.8) | 59.3 (+13.0) | 74.7 | 68.0 | 59.2 | 42.5 | 78.7 | 76.0 | 61.0 | 50.6 |
> | 4 | ERM | 65.0 | 46.5 | 65.0 | 54.0 | 71.4 | 41.1 | 72.5 | 58.0 | 51.1 | 32.9 |
> |  | PEARL | 67.2 (+3.4) | 52.5 (+13.0) | 71.4 | 60.0 | 60.7 | 38.9 | 75.9 | 66.0 | 60.8 | 45.2 |
> | 5 | ERM | 64.3 | 46.3 | 65.9 | 54.0 | 73.4 | 48.3 | 65.6 | 50.0 | 52.3 | 32.9 |
> |  | PEARL | 66.3 (+3.1) | 51.0 (+10.2) | 70.3 | 60.0 | 63.4 | 43.6 | 71.3 | 60.0 | 60.2 | 40.4 |
>
>
> ### Response to W2: Experiments on Many-shot Setting
>
> Thank you for your suggestion. In response, we expanded our evaluations to 8, 16，32, and 64 shots. We were unable to test with 128 shots as most sequences exceeded the maximum sequence length of 8k for Llama3. The results for average and worst-case performance gains are reported in Table 5 below.
>
> Table 5: Scaling to many-shot ICL.
> | Shots | Average Gain (%) | Worst Gain (%) |
> | :---: | :--------------: | :------------: |
> | 8     | 7.6              | 39.2           |
> | 16    | 5.3              | 23.7           |
> | 32    | 3.8              | 36.4           |
> | 64    | 3.5              | 36.7           |
>
> As shown in Table 5, our method, PEARL, achieves notable worst-case performance gains ranging from 24% to 40% when generalizing to larger shot numbers (up to 64 shots) and longer sequences (up to 8k). This indicates that PEARL helps LLMs learn robust features that generalize well to many-shot ICL settings.
>
> Detailed results and visualizations have been updated in Section 6.2 and Appendix G of our manuscript, and these updates are distinctly marked in orange for easy identification.
>
> We hope these additional results substantiate the generalizability of our method and adequately address your concerns.
>
>
> ### Response to W3: Explanation About Optimal Transport
>
> Thank you for your question. We provide an alternative interpretation of our modeling approach from a graph-theoretic view. The OT modeling implemented by the P-Net can be understood in two parts:
>
> 1. *Parametric part*: This acts as a feature extractor that takes $n$ demonstrations, producing their sentence representations $H \in \mathbb{R}^{n \times h}$, where $h$ is the dimensionality of the representations. Pairwise relationships among the demonstrations are modeled using a layer of cross-demonstration interaction:
>
>    $\mathbf{R}=g\left(H W H^{\top}\right)$,
>    where $W$ is a $d \times d$ weight matrix, and $g$ is a nonlinear activation function. The output matrix $\mathbf{R} \in \mathbb{R}^{n \times n}$ can be interpreted as an *adjacency matrix*. In this *graph-theoretic view*, the nodes represent demonstrations, and the relationship between nodes $i$ and $j$ is denoted by the edge $R_{ij}$. Here, we define $R_{ij}$ as the potential increase in difficulty for the LLM if demonstrations $i$ and $j$ are swapped; a higher value of $R_{ij}$ indicates that swapping these two demonstrations may significantly increase the task's difficulty. However, the current gap is that $R$ is not yet a probability distribution.
>
> 2. *Non-parametric part*: We employ the Sinkhorn algorithm to transform $\mathbf{R}$ into a probability distribution. This is achieved by normalizing each row and column iteratively until the matrix $\mathbf{R}$ approximately converges to a doubly stochastic matrix or the sinkhorn distribution. This corresponds to Equations (11-12) in our paper.
> Following this, we would like to sample a permutation matrix from this distribution, and the permutation matrix indicates which elements among the $n$ demonstrations should be swapped—entry '1' indicates a swap, and '0' indicates no swap. The sampling method used is Gumbel sampling, as described in Equation (13).
>
> Together, parts (1) and (2) comprise the entirety of P-Net. This allows the process of permutation mapping (from the input permutation to the target permutation) to be fully differentiable.
>
> To further ensure that the outputing permutation matrix presents a challenge to the LLM, we conduct adversarial training between P-Net and the LLM (Section 4.3).
>
> We have updated this explanation in Section 4.2 OT of our manuscript, hoping it will aid in better understanding the processes involved.

---

> ### Author Response · Authors · 2024-11-23
> **Rebuttal by Authors [3/3]**
>
> ### Response to W3: Formatting Corrections
> Thank you for your careful review and for pointing out the inaccuracies. We have updated Equation (1) and the formatting in the Algorithm section of our manuscript accordingly. Your meticulous feedback has undeniably enhanced the clarity of our presentation.
>
>
> ### Response to Q1: Clarification About Matrix W
> Thank you for your question. We apologize for any confusion caused by our unclear description initially. To clarify, the matrix $W$ mentioned is not a projection matrix, but rather a normal parameter matrix. Its function is to map the representations $H$ of the demonstrations into a new space for further interaction. We regret any confusion caused and have updated our manuscript accordingly.
>
> ### Response to Q2: Explanation Regards the Performance of ERM+DS/IM
>
> Thank you for your question. The decrease in average-case performance for the ERM+DS and ERM+IM baselines can be attributed to the trade-off between robustness and accuracy inherent in these methods.
>
> Instance Mixupintroduces new samples by linearly interpolating between existing examples. While this approach can enhance robustness by providing more varied training data and mitigating overfitting, it also introduces noise that can obscure meaningful patterns in the original data. This added noise may hinder the model's ability to learn crucial relationships, leading to decreased average-case performance. Demonstration Shuffling can be considered a gentler version of Instance Mixup, which is why its decline is smaller.
>
> In essence, both methods aim to improve robustness but may inadvertently compromise accuracy due to the introduction of noise (in the case of Mixup) or loss of beneficial ordering (in the case of Demonstration Shuffling). Our experimental results indicate that these simple learning strategies struggle to achieve an good balance between robustness and accuracy.
>
> We hope this explanation clarifies the observed performance of the ERM+DS and ERM+IM baselines. Please feel free to reach out with any further questions.
>
>
> ***
> Thank you once again for your insightful comments, which has undoubtedly strengthened our work. We have updated our manuscript to reflect these changes, which are highlighted in orange for easy identification. We hope that our responses could effectively address your concerns, and we look forward to  any further feedback you may have.

---

> > ### Comment · Reviewer_ZWPA · 2024-11-24
> > **Thank you for the extensive additional experiments**
> >
> > I would like to thank the authors for the extensive additional experiments. My main concerns, which were the lack of experiments for other models and of prompts with more than 5 shots, have been addressed. As such, I am raising my score to recommend acceptance.

---

> > > ### Author Response · Authors · 2024-11-25
> > > **Thank You for Your Feedback and Recognition**
> > >
> > > Thank you for your thoughtful and constructive feedback, as well as for recognizing our additional experiments. We greatly appreciate your time and effort in reviewing our work and are glad that our rebuttal addressed your concerns.

---

### Official Review · Reviewer_SLuu · 2024-11-04

**Soundness:** 3
**Presentation:** 3
**Contribution:** 2
**Rating:** 6
**Confidence:** 3

**Summary:**

The authors propose a method for improving the robustness of ICL to permutations in the demonstrations. The idea is to pose the problem as a min-max problem akin to distributionally robust optimization, where the LLM is trained to be robust over hard permutations generated by the adversarial distribution. The adversarial distribution is generated with another neural network and is learned jointly with the LLM, with alternating updates that are reminiscent of GANs. The authors present experiments that re-validate the existence of poor demonstration orders (using exhaustive search and search with their neural network), and demonstrate that their method, PEARL, outperforms two baselines (shuffling and mixup) on learning linear functions and instruction fine-tuning.

**Strengths:**

+ The paper combines distributionally robust optimization using transport maps with the min-max problem of learning over worst-case perturbations, appears to be a unique approach.
+ The attacker model for finding worst-case perturbation looks to have a favorable computational complexity.
+ The robustness approach improves upon random and mixup-based baselines.

**Weaknesses:**

Most of this feedback is centered around how this work is situated against / compares to existing work in the literature.

+ While the authors appear to be aware of some other works studying the fragility of ICL to demonstration order, I felt the paper did not situate their work relative to the existing studies. There is a non-trivial body of work on specifically studying demonstration ordering that has come out since the "Fantastically Ordered Prompts and Where to Find Them: Overcoming Few-Shot Prompt Order Sensitivity" paper by Lu et al. 2021, and this body feels like it should be discussed as a section in related work. Also related is Chen et al. 2023 (same problem but perhaps more theoretical in nature).
+ Similarly, the experiments do not compare how the approach compares to prior work in this space. For example, Chang & Jia (2022) quantify the influence of examples and their position and use this information to stabilize ICL.   Xiang et al. (2024) and Zhang et al. (2024) explicitly have methods that try to solve the permutation robustness problem for ICL.
+ Section 3 is dedicated toward "unveiling the vulnerability of LLMs to input permutations". The phrasing makes it sound like this is new contribution, but perturbation ordering at this point is not a new phenomenon.

Chang, Ting-Yun, and Robin Jia. "Data curation alone can stabilize in-context learning." arXiv preprint arXiv:2212.10378 (2022).
Chen, Yongqiang, et al. "Positional Information Matters for Invariant In-Context Learning: A Case Study of Simple Function Classes." arXiv preprint arXiv:2311.18194 (2023).
Zhang, Kaiyi, et al. "Batch-icl: Effective, efficient, and order-agnostic in-context learning." arXiv preprint arXiv:2401.06469 (2024).
Xiang, Yanzheng, et al. "Addressing Order Sensitivity of In-Context Demonstration Examples in Causal Language Models." arXiv preprint arXiv:2402.15637 (2024).

**Questions:**

+ Are the baselines in the paper supposed to correspond to methods explored in previous literature? I feel I've seen demonstration shuffling in other papers, and instance mixup suggests it is based on the mixup augmentation, but no citations or descriptions are given for either.
+ How novel is the attack method? Currently, I feel the contribution of the attack is not as strongly posed as it could be---there is some discussion on other approaches being exponential in cost, but no explicit experimental comparison that touts the concrete benefits. The current section just shows that the attack works, but the fact that it works is not surprising given the previous work. This could be a more significant result if compared more concretely with respect to the appropriate prior work.
+ Where do the authors see their work in this space, given the other methods for improving robustness to permutations and/or quantifying the impact of position? Are these comparable, or can they be discussed in any capacity? I would not personally require that these need to be added as additional rows in the experiment, as I fundamentally disagree with the "moving goalposts" that the field has become accustomed to. However, I do believe some amount of earnest and thoughtful discussion on how these approaches relate to each other would be fair to include, since the latest two were released in early 2024.

---

> ### Author Response · Authors · 2024-11-23
> **Rebuttal by Authors [1/3]**
>
> Thank you for your insightful and positive feedback. We appreciate your recognition of our method as a unique approach that combines distributionally robust optimization (DRO) using optimal transport (OT) with min-max learning to enhance the robustness of ICL against permutations. We are also pleased that you noted our attacker model's favorable computational complexity and the effectiveness of our robust learning method in outperforming baseline approaches. We will now address your comments sequentially and supplement our work with new experimental results and discussions.
>
> ## Response to W1: Related Work Discussion
>
> Thank you for pointing out the omission in our related work section. We have improved this section to better situate our work within the existing literature.
>
> Current research on order sensitivity in in-context learning can be divided into three categories: **structural improvements**,  **training-stage methods**, and **inference-stage methods**.
>
> Most **training-stage methods** focus on improving general performance in ICL [1,2] while neglecting robustness to permutations of demonstrations. Recent studies suggest that this phenomenon stems from the autoregressive nature of Transformer language models [3,4]. InfoAC [4] introduces contrastive learning during fine-tuning to break the autoregressive constraint and enable bidirectional token visibility; however, their approach achieves limited success and is restricted to classification tasks. Preliminary work by [3] seeks for **structural improvements** and they shows the DeepSet architecture exhibits better permutation invariance than the Transformer; however, this MLP-based new architecture is too small to solve complex language modeling tasks.
>
> **Inference-stage methods** can be categorized into four types:
>
>  - *Demonstration selection* [5,6], which primarily enhances normal-case performance without guaranteeing worst-case performance under permutations;
>  - *Output calibration* [7-9], which proves effective for classification tasks but is less applicable to generation tasks due to sequence calibration challenges;
>  - *Order optimization* [10], which aims to find the best ordering during inference but suffers from exponential computational complexity;
>  - *Prediction ensembling* [11], a recent work proposes transforming an $n$-shot ICL into $n$ one-shot predictions and ensembling the results; while effective for classification, this approach leads to decreased performance on generation tasks.
>
> In summary, inference-stage methods aim to circumvent order sensitivity by pre-/post-processing without fundamentally enhancing the robustness of LLMs to different orders. Moreover, most methods are designed for classification tasks and show reduced effectiveness on generation tasks.
>
> To the best of our knowledge, our work is the first to address this problem from an adversarial perspective. We propose a novel distributionally robust optimization (DRO)-based learning algorithm to enhance the inherent robustness of LLMs against order perturbations, and we solve it using the Sinkhorn operator. Our approach complements existing inference-stage methods and generalizes across diverse task categories.
>
> We have updated the related work section in our manuscript accordingly. We hope this addresses your concern, and we appreciate your suggestion, which has helped us improve the rigor of our work.
>
> Reference:
>
> [1] Min et al., 2022, MetaICL: Learning to Learn In Context
>
> [2] Wei et al., 2023, Symbol tuning improves in-context learning in language models
>
> [3] Chen et al., 2023, Positional Information Matters for Invariant In-Context Learning: A Case Study of Simple Function Classes
>
> [4] Xiang et al., 2024, Addressing Order Sensitivity of In-Context Demonstration Examples in Causal Language Models
>
> [5] Chang et al., 2023, Data Curation Alone Can Stabilize In-context Learning
>
> [6] Peng et al., 2024, Revisiting Demonstration Selection Strategies in In-Context Learning
>
> [7] Zhao et al., 2021, Calibrate Before Use: Improving Few-shot Performance of Language Models
>
> [8] Li et al., 2023, Distinguishability Calibration to In-Context Learning
>
> [9] Guo et al., 2024, What Makes a Good Order of Examples in In-Context Learning
>
> [10] Lu et al., 2022, Fantastically Ordered Prompts and Where to Find Them: Overcoming Few-Shot Prompt Order Sensitivity
>
> [11] Zhang et al., 2024, Batch-ICL: Effective, Efficient, and Order-Agnostic In-Context Learning

---

> ### Author Response · Authors · 2024-11-23
> **Rebuttal by Authors [2/3]**
>
> ## Response to W2:  Baseline Comparisons
>
> Thank you for highlighting these important baselines. We have added comparisons with these methods in our updated manuscript. Since CurStable ([5] Chang et al., 2023) and Batch-ICL ([11] Zhang et al., 2024) are inference-based methods and are complementary to our approach, we have not only compared our method with them but also considered combining our method with theirs. The new results are presented in Table 1 (also updated in Table 2 in the manuscript).
>
> Our comparisons show that PEARL achieves superior performance compared to both training-stage and inference-stage methods. Among inference-stage methods, Batch-ICL boosts both average and worst-case performance on classification tasks (CSQA, CoLA); however, it exhibits limited or negative effects on generation tasks (CurDial, TMW), limiting its applicability. In contrast, CurStable, through demonstration selection, performs well on both task types. Moreover, combining PEARL with inference-stage methods further improves performance.
>
> We hope that these additional baseline comparisons effectively address your concern.
>
> Table 1: Average and worst-case performance with added baselines. Performance gain (%) are in parentheses.
> | # Shot | Method | Average | Average | CSQA | CSQA | CurDial | CurDial | CoLA | CoLA | TMW | TMW |
> | :---: | :---: | :---: | :---: | :---: | :---: | :---: | :---: | :---: | :---: | :---: | :---: |
> |  |  | Avg. | Worst. | Avg. | Worst. | Avg. | Worst. | Avg. | Worst. | Avg. | Worst. |
> | 2 | InfoAC | 55.7 (-2.9) | 47.6 (-3.7) | 57.2 | 55.0 | 53.4 | 36.4 | 63.3 | 61.5 | 48.7 | 37.3 |
> |  | CurStable | 61.6 (+7.5) | 52.1 (+5.4) | 64.0 | 56.0 | 61.7 | 46.2 | 68.4 | 62.0 | 52.3 | 44.1 |
> |  | Batch-ICL | 58.6 (+2.2) | - | 63.1 | - | 56.3 | - | 65.3 | - | 49.6 | - |
> |  | Pearl | 62.9 (+9.8) | 56.4 (+14.2) | 65.0 | 62.0 | 60.3 | 50.7 | 71.0 | 68.0 | 55.1 | 44.8 |
> |  | Pearl + CurStable | 65.6 (+14.5) | $58.0(+17.4)$ | 68.0 | 63.0 | 64.6 | 52.8 | 74.0 | 70.0 | 55.9 | 46.2 |
> |  | Pearl + Batch-ICL | - | - | 65.5 | - | - | - | 72.2 | - | - | - |
> | 3 | InfoAC | 56.3 (-2.6) | 39.5 (+3.1) | 59.4 | 49.0 | 55.2 | 24.3 | 62.1 | 55.8 | 48.4 | 28.8 |
> |  | Curstable | 61.0 (+5.4) | 41.4 (+8.0) | 65.0 | 52.0 | 62.5 | 26.7 | 64.0 | 54.0 | 52.3 | 32.7 |
> |  | BATCH-ICL | 58.6 (+1.3) | - | 62.0 | - | 59.6 | - | 64.0 | - | 48.7 | - |
> |  | Pearl | 63.1 (+9.2) | 46.9 (+22.5) | 68.4 | 62.0 | 66.7 | 34.8 | 64.7 | 56.0 | 52.4 | 34.7 |
> |  | Pearl + CurStable | 65.0 (+12.5) | 48.9 (+27.5) | 70.0 | 64.0 | 67.6 | 35.8 | 68.4 | 58.0 | 54.1 | 37.6 |
> |  | Pearl + Batch-ICL | - | - | 68.4 | - | - | - | 65.6 | - | - | - |
> | 4 | InpoAC | 58.6 (-1.8) | 33.0 (+7.8) | 63.7 | 44.6 | 58.7 | 19.0 | 63.9 | 51.3 | 48.1 | 17.0 |
> |  | CurStable | 60.8 (+1.8) | 32.3 (+5.6) | 63.0 | 40.0 | 64.5 | 22.8 | 64.1 | 48.0 | 51.5 | 18.4 |
> |  | BATCH-ICL | 58.5 (-2.0) | (5.6). | 62.0 | - | 61.5 | - | 63.3 | - | 47.2 | - |
> |  | Pearl |  |  | 68.4 | 52.0 | 69.2 | 31.3 | 64.7 | 52.0 | 50.1 | 23.0 |
> |  | Pearl + Curstable | 65.0(+8.8) | 41.4 (+35.1) | 70.6 | 54.0 | 72.3 | 34.2 | 66.3 | 54.0 | 50.6 | 23.2 |
> |  | Pearl + Batch-ICL | - | - | 69.0 | - | - | - | 65.0 | - | - | - |
>
> ## Response to W3: Clarification of The Contribution
>
> Thank you for your insightful comment. We apologize for any confusion caused by our phrasing. In Section 3, our goal is not to introduce the vulnerability of LLMs to input permutations as a new discovery but to revisit and empirically assess this issue in the context of current SOTA open-source models. We focus on validating the two questions to motivate our research:
>
> 1.  *Has order sensitivity been resolved in latest SOTA open-source LLMs like Llama 3? If not, what is its severity?*
> 2.  *Can this vulnerability be exploited as a new means of attack to perturb LLMs, and how effective could such attacks be?*
>
> By investigating these questions, we aim to highlight the ongoing significance of order sensitivity in modern LLMs, underscoring the necessity for robust solutions. This examination sets the stage for our main contribution: proposing a novel method to enhance the robustness of LLMs against input permutation perturbations.
>
> We have also revised the phrasing in the manu and we hope it can address your concerns.
>
> ## Response to Q1:  Missing Citations
> Thank you for noting these missing citations. We have added it to the manuscript.

---

> ### Author Response · Authors · 2024-11-23
> **Rebuttal by Authors [3/3]**
>
> ## Response to Q2:  Novel of Attack Methods and Comparative Analysis
> Thank you for your thoughtful comment. We acknowledge that this concern may also relate to Weakness 3. In our revisiting section, we seek to explore adversarial attacks on ICL based on permutations, thereby introducing this new attack scenario. To the best of our knowledge, there are currently no existing attack methods specifically designed for the proposed permutation-based attack scenario.
>
> Existing adversarial attacks on prompts and ICL mainly fall into two categories:
>
> 1.  Prefix/Suffix Optimization: Methods that optimize a prefix or suffix to explicitly prepend or append to the prompt [12,13,14].
>
> 2.  Noise Perturbation in Samples: Methods that introduce noise perturbations into the ICL demonstration samples [15].
>
>
> However, these methods cannot be directly applied to permutation-based attack scenarios because learning to generate permutations that can maximize adversarial impact is a non-trivial problem. Besides exhaustive search, we propose a neural-based attack that utilizes a neural network equipped with optimal transport (OT) to learn adversarial permutations.
>
> Our assessment of the severity of attacks in this new scenario is to highlight the need for robust solutions. This motivates our main contribution: introducing a DRO-based learning algorithm to enhance the inherent robustness of LLM’s ICL abilities.
>
>
> We hope that this explanation can address your concerns, and we look forward to any further feedback you may have.
>
>
> Reference
>
> [12] Universal and Transferable Adversarial Attacks on Aligned Language Models. 2023
>
> [13] Tricking LLMs into Disobedience: Understanding, Analyzing, and Preventing Jailbreaks 2023
>
> [14] Hijacking Large Language Models via Adversarial In-Context Learning. 2023
>
> [15] Data Poisoning for In-context Learning. 2024
>
>
> ## Response to Q3:  Discussion and Comparison with Recent Baselines
> Thank you for raising this point. We have incorporated them into the responses to Weaknesses 1 and 2, as well as into the updated manu. We hope this can address your questions.
>
>
> ***
> Thank you once again for your insight comments, which has undoubtedly strengthened our work. We have updated our manuscript to reflect these changes, which are highlighted in orange for easy identification. We hope that our responses could effectively address your concerns, and we look forward to  any further feedback you may have.

---

### Author Response · Authors · 2024-11-24
**General Response**

Dear AC and Reviewers,

We sincerely appreciate your time and expertise in reviewing our paper. We are pleased that you found our work to be unique and elegant (ZWPA, cDtR, SLuu), particularly in our design of the P-Net (threat model) which employs Optimal Transport (OT) and the Sinkhorn algorithm to learn challenging permutations for LLMs. The approach is recognized as both technically elegant and practical for practitioners in their fine-tuning pipelines (ZWPA), with favorable computational complexity and no additional inference-time overhead (ZWPA, SLuu). We are glad that our Distributionally Robust Optimization (DRO)-based training algorithm, working in concert with P-Net, effectively improves LLMs' robustness to ICL permutations (ZWPA, cDtR, n5jf), as demonstrated through our comprehensive experimental evaluation (n5jf). Additionally, we are pleased that the clarity and presentation of our manuscript were well-received (ZWPA, cDtR, n5jf).


We believe that all of the reviewers' concerns can be addressed. Below, we provide brief responses to the main concerns and suggestions raised in the reviews:

**Comparison With Baselines** (SLuu, cDtR, n5jf): We have extended our discussions and comparisons to include four additional baselines: three inference-time baselines and one training-phase baseline. Our method demonstrates substantial improvements over these baselines, achieving 9–21% performance gains. Furthermore, when combined with inference-time methods, our approach yields additional performance improvements of 3–5%, highlighting the complementary nature of our method. We have also updated the related work section to better situate our contributions within the existing literature (*see Section 2 and Table 2 in Section 6*).

**Experiments on Diverse LLMs** (ZWPA, n5jf): To validate the general applicability of our method, we have expanded our experiments to include four additional LLMs: Mistral 7B, Gemma 7B, Llama 2 7B, and Llama 2 13B. The results, provided in the rebuttal and updated manuscript, demonstrate consistent and significant performance gains ranging from 7% to 59% when the number of shots is greater than 2. These improvements across diverse model families indicate the broad applicability of our approach across different LLMs (*see Tables 6–9 in Appendix G*).

**Scaling to Many-Shot Settings** (ZWPA, cDtR): We have extended our evaluations to 8-, 16-, 32-, and 64-shot settings. The results show that our method generalizes well to large-shot and long-sequence scenarios, achieving an impressive 24–40% performance gain, even when trained with smaller settings. This demonstrates the excellent generalization capability of our approach (*see Figure 5 in Section 6 and Table 10 in Appendix F*).

**Clarification on Threat Model and Training Process** (n5jf): We acknowledge that our initial wording may have led to misunderstandings regarding the attack setting and training process of P-Net and the LLM. To address this, we have carefully revised the formulas and explanatory text in Sections 3 and 4 of the manuscript to enhance clarity and make it easier to understand. Additionally, we have provided explanations of these points in our responses (*see Sections 3 and 4, revisions highlighted in blue*).

All these revisions have been made in the manuscript and are highlighted for your convenience.

Please let us know if there are any outstanding concerns, and we are happy to discuss them further. We would appreciate it if you could take our responses into consideration when making the final evaluation of our work.

---

### Meta-Review · Area_Chair_NvFS · 2024-12-20

**Metareview:**

The submission proposes a method for improving the robustness of in context learning to permutations in the demonstrations.

+ The paper is well-written.
+ The method is technically sound.

- There were some concerns raised regarding limited empirical evaluation.

**Additional Comments On Reviewer Discussion:**

There were some issues raised in the initial reviews, including
- comparisons with baselines
- using diverse LLMs in the experiments
- scaling to many-shot setting
- technical clarifications on the threat model and training procedure.
The rebuttal was carefully considered by the reviewers, who have acknowledged that the issues they raised have been sufficiently addressed. While one of the scores is still negative, this reflects the pre-rebuttal state. I believe that the thorough rebuttal provided by the authors has answered the questions raised. Hence, despite the mixed ratings, in my opinion the paper is ready for publication.

---

### Decision · Program_Chairs · 2025-01-22

Accept (Poster)